# Long-Lasting Mucosal and Systemic Immunity against Influenza A Virus Is Significantly Prolonged and Protective by Nasal Whole Influenza Immunization with Mucosal Adjuvant N3 and DNA-Plasmid Expressing Flagellin in Aging In- and Outbred Mice

**DOI:** 10.3390/vaccines7030064

**Published:** 2019-07-16

**Authors:** Jorma Hinkula, Sanna Nyström, Claudia Devito, Andreas Bråve, Steven E. Applequist

**Affiliations:** 1Division of Molecular Virology, Department of Clinical and Experimental Medicine, University of Linköping, SE-581 83 Linköping, Sweden; 2Center for Infectious Medicine, F59, Department of Medicine, Karolinska Institutet, Karolinska University Hospital Huddinge, SE-141 86 Stockholm, Sweden; 3Division of Immunology, HD Immunity, 126 25 Stockholm, Sweden; 4Public Health Agency of Sweden, 171 82 Stockholm, Sweden

**Keywords:** influenza, immunization, intranasal, adjuvant, lipid, flagellin

## Abstract

*Background*: Vaccination is commonly used to prevent and control influenza infection in humans. However, improvements in the ease of delivery and strength of immunogenicity could markedly improve herd immunity. The aim of this pre-clinical study is to test the potential improvements to existing intranasal delivery of formalin-inactivated whole Influenza A vaccines (WIV) by formulation with a cationic lipid-based adjuvant (N3). Additionally, we combined WIV and N3 with a DNA-encoded TLR5 agonist secreted flagellin (pFliC(-gly)) as an adjuvant, as this adjuvant has previously been shown to improve the effectiveness of plasmid-encoded DNA antigens. *Methods*: Outbred and inbred mouse strains were intranasally immunized with unadjuvanted WIV A/H1N1/SI 2006 or WIV that was formulated with N3 alone. Additional groups were immunized with WIV and N3 adjuvant combined with pFliC(-gly). Homo and heterotypic humoral anti-WIV immune responses were assayed from serum and lung by ELISA and hemagglutination inhibition assay. Homo and heterotypic cellular immune responses to WIV and Influenza A NP were also determined. *Results*: WIV combined with N3 lipid adjuvant the pFliC(-gly) significantly increased homotypic influenza specific serum antibody responses (>200-fold), increased the IgG2 responses, indicating a mixed Th1/Th2-type immunity, and increased the HAI-titer (>100-fold). Enhanced cell-mediated IFNγ secreting influenza directed CD4^+^ and CD8^+^ T cell responses (>40-fold) to homotypic and heterosubtypic influenza A virus and peptides. Long-term and protective immunity was obtained. *Conclusions*: These results indicate that inactivated influenza virus that was formulated with N3 cationic adjuvant significantly enhanced broad systemic and mucosal influenza specific immune responses. These responses were broadened and further increased by incorporating DNA plasmids encoding FliC from *S. typhimurum* as an adjuvant providing long lasting protection against heterologous Influenza A/H1N1/CA09pdm virus challenge.

## 1. Introduction

250,000–500,000 people die from influenza or bacterial infections every year following influenza infection (www.who.int/influenza/en/). Viral spread also results in considerable days of illness and the loss of millions of work days annually. Influenza A virus is an RNA virus with a segmented genome of eight genes. The two surface proteins hemagglutinin (HA) and neuraminidase (NA) are the main targets for the neutralizing antibodies. The combination of these two antigens (20 different serotypes of HA [HA1 to HA20] and eleven NA in [NA1 to NA11]) with the two most recent bat influenzas identified, greatly determines the variability between the influenza virus strains [1]. Human vaccination using these immunodominant antigens is a primary method of influenza prevention that is used to control both seasonal and pandemic influenza strains [2]. When unchecked, seasonal and pandemic influenza both strongly affect the elderly who are especially sensitive to complications following influenza infection. Furthermore, existing influenza vaccines are less effective in the elderly when compared to younger people. The development of mucosally administered live or killed inactivated adjuvanted vaccines would be one way to create vaccines that are more conveniently delivered efficiently to the elderly [3].

It would be highly desirable to develop influenza vaccines that provide broader influenza-specific immune responses than what can be obtained with the currently available commercial inactivated flu-vaccines. If stronger and more long-lasting, cell-mediated and humoral flu-specific immunity could be obtained, it would be more likely that the obtained immunity could better protect against disease in future epidemics. In preclinical models, it has been reported that the killed formalin-inactivated influenza vaccines nasally given induce immunity almost equally with or without adjuvants [4,5,6]. Nevertheless, it would be advantageous to broaden the often elicited homosubtypic immunity into a heterosubtypic immune response recognizing more divergent influenza virus strains. Several recent preclinical studies in mice suggest that this is possible [7,8,9,10] by using virus-like particle vaccines [11]. It seems clear that both arms of the humoral adaptive immune system will need to be employed to broaden vaccine immunity, with influenza A neutralizing serum IgG, and also preferably with mucosal immunity consisting of secretory IgA towards the outer envelope proteins HA and NA [12]. This should then be combined with a systemic cell-mediated immunity as the second line of defense, which consists of CD8+ T cells recognizing conserved internal influenza virus epitopes [13], as well as a broad repertoire of memory CD4+ Th cells, which are critical to the maintenance of long-lasting humoral and CD8+ T cell immunity [14,15,16].

A new pandemic would probably be more rapidly spread and extensive than the Spanish flu of 1918–1919 when considering growing human populations and ease of international travel [17]. Indeed, these factors appear to have facilitated the emergence of the recent A/H1N1/“Swine” influenza, which appears to be a mixture of influenza viruses previously not seen in man, such as three triple reassorted genes from north American swine and human, three genes from classical swine influenza, and two genes from Eurasian swine [18]. When another pandemic appears, there will be many challenges to overcome in order to rapidly develop an effective vaccine against influenza, especially if they evolve from complex reassorted gene mixtures, as seen with the 2009 Swine-origin 2009 A (H1N1) influenza viruses. The time it takes to produce influenza vaccines needs to be decreased as well as issues of immunogenicity, quality control, and safety. Development and testing of vaccines while using new technologies are to be lauded. However, they can introduce unquantified risk in the development chain slowing vaccine development. Finding ways to improve upon existing technologies may be a way of mitigating development risk, while at the same time improving immunogenicity, safety, and production speed. The aim of this study was to use a licensed, existing whole formalin-inactivated influenza A virus (WIV) as a source of antigen and improve upon its ability to elicit immune responses by the addition of adjuvants. WIV vaccines are well known to induce poor cellular immune responses, unless combined with adjuvants [19]. This study investigated how additon lipid and genetic adjuvant(s) could be formulated to contain several critical components that cooperate to provide both a strong humoral, systemic and mucosal, as well as systemic cell-mediated heterosubtypic immune response, in both inbred mice (C57BL/6) and the outbred NMRI mice. Flagellin is an agonist of TLR5, but it is also directly recognized by the cytosolic Nod-like receptor family member Naip5, which signals through NLRC4 to form an inflammasome. Soluble flagellin has been shown to be a potent adjuvant in numerous studies and triggers numerous immune responses [20]. In previous studies, the presence and uptake of the bacterial flagellin proteins by CD103+ dendritic cells (DC) have resulted in their increased presence in mesenteric lymphnodes. Further, the flagellin-proteins have been shown to increase B-cells to switsch into IgA secreting cells, thereby enhancing the mucosal B and T cell responses against antigenic proteins [21]. However, there are few studies on the adjuvant effects of DNA-encoded flagellin [19]. The novelty of the present vaccination design of inactivated influenza A virus is the combination of previously never studied combined adjuvants. Thus, the adjuvants, the cationic lipid N3 alone, or N3 lipid mixed with DNA-plasmid expressing the TLR5-agonistic, de-glycosylated flagellin C-protein mixed with the WIV/Salomon Island/2006 A/H1N1-antigen prepared as an emulsion for nasal mucosal administration in two strains of aging mice.

The reasons for proposing mucosal administration would be to obtain mucosal immunity in the nasal and respiratory organs, where respiratory viral infections enter the body, to provide mucosal first-line barrier immunity. Furthermore, the immunization protocol aimed to study the long-term protective effect in an aging target-group of elderly individuals of mice reaching 22–23 months of age (representing human ages 60–70 yrs), where the influenza specific systemic and mucosal Th1/Th2-type immune pattern responses were followed and they could be correlated with different vaccination designs and levels of heterologous influenza A virus protection.

Here, we evaluated the potential benefits of combining the WIV vaccine with an experimental mucosal cationic oleic oil-based adjuvant alone and with a DNA-plasmid expressing a secreted form of flagellin from *Salmonella typhimurium*.

## 2. Materials and Methods

### 2.1. Inactivated Viral Vaccine, Lipid Adjuvants, and Formulations

Whole inactivated viral (WIV) vaccines are whole viral particles that are inactivated with formalin [22]. The influenza strain A/H1N1/Salomon Island/2006 (A/H1N1/SI) was used as a model vaccine candidate. The adjuvant N3 is based on a natural human fatty acid (L3) that is composed of: oleic acid 92%, linoleic acid 6%, and saturated monoolein 2% [23], and modified by coupling an amine-group to obtain a charged cationic molecule, N3 [24].

### 2.2. DNA Expression Adjuvant Constructs and Immunizations

pFliC-Tm(-gly) *S. typhimurium* has been described previously [25]. pFliC-Tm(-gly) was subjected to site-directed mutagenesis to insert two in-frame translational stop-codons after AA 459 of FliC(-gly) to generate a secreted version of FliC(-gly) (AA numbering is based on GenBank Accession #D13689). Changes were confirmed by DNA sequencing. The immunizations were intranasaly performed (i.n.) as previously described (Table 1) with five groups of outbred female NMRI mice and five groups of inbred female C57BL/6J mice. Briefly, mice were sedated with isoflurane (4% in air) and given WIV vaccine 5 µL/nostril (total volume 10 µL/mouse, 1.5 µg HA antigen/mouse or a total of 25 µg protein/mouse). In groups where the N3 adjuvant was used, a 1% concentration was intranasally given, 6 uL/nostril. Female mice of the NMRI strain were purchased from ScanBur, Sollentuna, Sweden, and C57BL/6 mice were purchased from Charles River, Dortmund, Germany. The animals were kept until 13 (±1) months before immunizations were initiated in accordance with ethical guidelines and permissions at the AF animal facility at the Karolinska Institutet, Stockholm, Sweden under specific pathogen free conditions [26].

### 2.3. ELISA Detection of Anti-Influenza A IgG, IgG isotypes, and IgA

IgG and ELISA measured IgA responses to influenza A in samples, as described [26]. The plates were coated with inactivated influenza A antigen (Swedish Institute for Communicable Disease Control, Solna, Sweden and Solvay Pharmaceuticals, BV, Weesp, Holland and recombinant HA/influenza A/H1N1/CA09pdm or NP Protein BioSciences, CT, USA) that was diluted to 2 µg/mL in sodium carbonate buffer pH 9.5–9.7 before 100 µL was added to each well. Influenza A positive mouse serum and naïve mouse serum were used as the controls for mouse anti-influenza A reactivity. The coated plates were washed with phosphate buffer saline (PBS)/0.05% Tween 20 (Sigma-Aldrich, S:t Louis, MO, USA) and then blocked with PBS/5% dry milk at 37 °C for 1 h followed by one wash. Mouse sera was diluted in PBS (pH 7.4)/0.5% bovine serum albumine (BSA, Boehring Mannheim, Mannheim, Germany)/0.05% Tween 20, and 100 µL of serial dilutions (1/50–1/5,000,000) were added to each well and then incubated at 37 °C for 90 min. After incubation, the plates were washed and 100 µL of HRP-conjugated goat-anti mouse IgG (BioRad, Richmond, VA, USA) or HRP-conjugated anti-mouse IgA (Southern Biotechnologies, Birmingham, AL, USA) (1:1000) diluted in 2.5% dry milk/0.05% Tween 20 (1:2000) was added to each well. The plate was incubated for 1 h at 37 °C and then washed. Ortho-phenylene diamine (OPD, Sigma) substrate was prepared by solving OPD-tablets 2 mg/mL in 0.1 M citrate buffer/0.003% H_2_O_2_. 100 µL was added to each well and the plate was then covered and incubated at room temperature for 30 min. The reaction was stopped by adding 100 µL 2.5M H_2_SO_4_ to each well and the absorbance was measured at OD 490 nm (24). The avidity index (AI) was determined by using the 8M urea wash procedure against the influenza antigens. IgG isotype reactivity to WIV was tested while using the ISO-2 ELISA reagent kit (Sigma), as recommended by the manufacturer. Isotype calculations of IgG1/IgG2a or 2c-ratios were calculated by dividing the OD 490 nm values for each subclass at dilution 1/100 or 1/1000. Inter-group ratio comparisons were made while using unpaired two-tailed, student t test. The ratio comparisons within each group were made using Pearsons correlation coefficient r.

### 2.4. Total IgA Quantification and Detection of Lung Anti-Influenza A IgA Responses

Lung-washes were harvested by flushing the lungs with PBS that was supplemented with protease inhibitors (Complete Mini, Roche, Mannheim, Germany) and then subjected to total IgA isolation while using the Kaptive IgA/IgE reagents (Biotech IgG, Copenhagen, Denmark) as recommended by the manufacturer. Total isolated IgA quantities were determined using an in-house murine IgA capture ELISA. Briefly, purified lung-wash IgA and standard mouse IgA (1 mg/mL, Sigma) was diluted ten-fold (PBS/5% dry-milk/0.05% Tween 20). 100 µL/per dilution was added to a 96-microwell plate that was precoated with rabbit anti-murine IgA (Dakopatts AB, Copenhagen, Denmark) and then incubated at 37 °C for 1 h. The plates were washed four times with PBS/0.05% Tween 20 before 100 µL HRP-conjugated goat anti-murine IgA was added to each well (Southern Biotechnologies) (1:1000). After 1 h incubation at 37 °C plates were washed and bound conjugate was detected by using OPD, as described above. The reactions were terminated using 100 µL/well 2.5M H_2_SO_4_ and the absorbance was measured at OD 490 nm. Total IgA was determined by comparing the OD-values of the test samples with the IgA standard. Detection of anti-influenza A IgA in total lung-wash IgA was done as with IgA from serum (above).

### 2.5. Hemagglutination Inhibition Assay

The hemagglutination inhibition assay (HAI) was used to quantify the antibodies against viral influenza A particles in serum from individual mice, as described previously [27]. An HI titre ≥ 40 was defined as a protective amount of serum antibodies [28]. Briefly, serum from individual mice were treated with receptor-destroying enzyme (RDE) overnight at 37 °C to remove the non-specific serum HAI inhibitors [29]. RDE was inactivated by incubation at 56 °C for 30 min., followed by the addition of 350 µL NaCl 0.9%. The HAI assay was initiated by adding 25 µL PBS to each well of a microtitre plate, followed by the addition of 50 µL of RDE treated serum. Serum was diluted in two-fold serial dilutions. 25 µL of influenza A/H1N1/SI or A/H1N1/CA09pdm containing four haemagglutinating units (HU) was added to each well. The plate was shaken, covered, and incubated at 20–25 °C for 15 min. Subsequently, 50 µL guinea pig erythrocytes were added, mixed, and followed by incubation for one hour at 4 °C. Thereafter, the plate was evaluated for hemagglutination and the degree of hemagglutination inhibition (HAI).

### 2.6. T-Cell Responses to Influenza A Antigens

T cell analysis was performed, as described here. Briefly, the splenocytes were isolated by physical disruption of the spleens, followed by Ficoll purification and washed twice with PBS. The depletion of CD8^+^ T cells was performed while using Dynabeads (Dynal Biotech, Oslo, Norway), according to the manufacturer’s instructions. The efficiency of CD8^+^ cell depletion was confirmed by flow cytometry. On average, 98% ± 2.4% of the CD8^+^ cells were removed. Total and CD8+ T cell depleted splenocytes from individual animals were suspended in RPMI 1640 (Sigma) supplemented with penicillin/streptomycin (Invitrogen) and 10% fetal calf serum (FCS, Sigma) and then subjected to anti-Interferon-γ (IFNγ) (Mabtech, Nacka, Sweden) antibody coated 96-well polyvinylidene fluoride (PVDF) bottomed plates (MAIPN 4510, Millipore Corporation, Bedford, MA, USA). WIV antigen restimulation was performed while using influenza A/H1N1/2006/SI or A/H3N2/1995/Wuhan at 100 TCID50. Peptide restimulations were performed using the H-2Kd binding NP peptides TYQRTRALV (147-156/aa), RLIQNSLTIERMVLS (55-69/aa) and the H-1Kb binding NP peptide ASNENMDAM (366-374/aa) at 1 µM final concentration (GenScript, Piscataway, NJ, USA). Concanavalin A (1 µg/well, Sigma) was used as a positive control to test cell activation and medium alone was used as the negative control. Spot-forming cells were quantified after 24 h incubation and then counted by an AID ELISPOT reader (AutoImmun Diagnostika, Strassberg, Germany). The results are given as cytokine-producing spot-forming cells (SFC) per million plated cells. The total and CD8^+^ depleted splenocytes (10^6^) were stained for 30 min. at 4 °C with FITC conjugated anti CD4 antibodies and with PerCP conjugated anti-CD8α antibodies (BD Pharmingen, Stockholm, Sweden). IL-5 production was determined by ELISA after restimulation of total splenocytes from individual mice with WIV A/H1N1/SI (1 µg total), as determined by the manufacturer (Omninvest, Budapest, Hungary) after 48 h from C57BL/6 samples and 72 h from the NMRI samples. The different time points chosen for the two mouse strains were determined in an in vitro pre-study influenza/ConA stimulation of spleen cells, and the optimal time point for highest levels of IL-5 secretion in elderly animals was chosen. Furthermore, IL-5 secretion was shown to be secreted at higher amounts for a longer period than IL-4 in vitro (or possibly consumed less rapidly) in aged mice, thus making IL-5 easier to use as a Th2-biomarker than IL-4, as shown by McDonald et al. 2017.

### 2.7. Influenza Challenge

To obtain information regarding tge longevity of heterologous influenza A-directed protective immunity mice were kept for up to 270 days after final booster immunization. Thereafter, at day 180 and at day 270, the influenza vaccinated and unvaccinated mice were intranasally challenged with influenza A/H1N1/2009pdm virus (10× LD50/mouse). Mice were monitored daily for four weeks post challenge, and when body weight loss was 20% or more the mice were sacrificed according with animal guidelines. Mice reaching this time point of sacrifice were sedated with isoflurane and blood and spleens were collected for the final immune analysis post-challenge.

### 2.8. Data Analysis and Statistics

Data was analyzed while using Prism v5.0d (GraphPad Inc.La Jolla, CA, USA). Confidence levels (95.0%) and the differences between the groups and doses of vaccines (nonparametric Mann–Whitney U test) (Table 2) were calculated using Prism. A significant difference was considered when a *p*-value of <0.05 was obtained. Inter-vaccination group ratio comparisons of IgG isotypes were made using unpaired two-tailed, student *t* test. IgG isotype ratio comparisons within each vaccination group were made while using Pearsons correlation coefficient r.

## 3. Results

### 3.1. Serum IgG and IgA Antibody Responses

The anti-influenza serum IgG and IgA titers in NMRI mice were significantly higher than baseline in animals receiving WIV combined with N3 adjuvant or N3 combined with FliC-DNA adjuvant already at day 21 after a single immunization (Figure 1A,C).

WIV given with pFliC(-gly) DNA did not result in significantly higher serum IgG titers then seen in mice that were immunized with only WIV demonstrating the important contribution of N3 to the effect of pFliC(-gly). The highest serum IgG titers were seen when both N3 and pFliC-DNA were combined with WIV with 100–140-fold increased titers. Booster immunizations at day 60 resulted in further elevation of the IgG and IgA titers (Figure 1A,C). Three weeks after booster immunization, the serum IgG in non-adjuvanted WIV immunized animals increased two-fold and IgA-titers increased five-fold. However, with WIV immunizations containing N3, a 100-fold serum IgG and 50-fold IgA increase was seen over WIV alone. Similar increases were also observed in mice receiving WIV with N3 and pFliC(-gly). Thus, the humoral influenza-specific immune responses between mice receiving WIV with N3 or N3 and FliC-DNA were not significantly different when analyzed by ELISA. Thus, binding antibodies alone may provide a misleading immune pattern from a functional point of view, as can be seen in the more detailed assays, such as subclass IgG ELISA (Figure 2, Figure 3 and Figure 4) or functional antiviral assays, such as in virus-inhibition assays (Figure 5). IgG isotype comparisons revealed a correlation with the use of pFliC-DNA and the appearance of stronger IgG2a responses (Table 2).

The most pronounced effect on the IgG1/IgG2a ratio was seen after the primary immunization with WIV in animals receiving N3 with pFliC-DNA (*p* < 0.01) in contrast to WIV alone or WIV with only N3 or pFliC-DNA alone. Both the IgG1 and IgG2a serum titers increased after the booster immunization, but the IgG1 titers increased more than IgG2a titers. In all groups of influenza vaccine immunized mice receiving WIV with all adjuvants both subclasses IgG1 and IgG2a responses were seen, which indicated a mixed Th1/Th2 immune response. Nevertheless, the inclusion of pFliC(-gly) skewed the response away from Th2.

In C57BL/6 mice, significant serum IgG and IgA titer increase over baseline was only seen when the influenza vaccine and N3 adjuvant was used alone or when N3 was combined with pFliC(-gly) (Figure 1B,D). Similar to NMRI mice, when WIV was given with N3 or N3 and pFliC(-gly), a 10-fold and 20-40-fold increased serum IgG was observed, respectively. As observed in the NMRI mice, WIV given with pFliC(-gly) DNA did not result in significantly higher serum IgG titers than that seen in mice that were only immunized with WIV. After a booster-immunization serum, the IgG titers were doubled and IgA titers increased four-fold in vaccinated non-adjuvanted C57BL/6 mice. However, mice receiving booster WIV immunization with N3 increased the serum IgG titers 100-fold and 1000-fold when N3 was combined with pFliC(-gly) as compared to WIV alone. When compared to mice receiving WIV alone, serum IgA titers increased 20 to 100-fold with the use of N3 or N3 with pFliC(-gly). Similar to NMRI mice, C57BL/6 mice receiving WIV and pFliC(-gly) immunizations did not have significant increases in antigen-specific IgG or IgA responses at either day 21 or 90. IgG isotype comparisons revealed a higher Th1-like response in mice that were given WIV with N3 and pFliC-DNA (Table 2). Booster immunization induced a mixed Th1/Th2-type immune response with increases in the titer of both IgG subclasses. As with the NMRI mice, after the booster immunization both the IgG1 and IgG2c serum titers increased, but the IgG1 titers increased more than the IgG2c titers. The inclusion of pFliC(-gly) also skewed responses to a Th1-like IgG isotype.

### 3.2. Mucosal IgA and IgG Responses

Lung washes were collected after booster immunization to study the presence of influenza A specific immunoglobulins in the airways. Lung IgA and low levels of IgG specific for Influenza A were seen in all groups of immunized mice (Figure 3). The highest IgA titers were obtained in both mouse strains, where WIV was given together with N3 or N3 and pFliC(-gly) (Figure 3A).

No significant differences were seen between WIV immunized non-adjuvanted mice and mice receiving WIV with pFliC(-gly) alone. Similar trends were observed in the lung IgG titers (Figure 3B), and both subclasses IgG1 and IgG2a were detected. The highest lung IgA and IgG titers were seen in the groups where WIV was given with N3 and pFliC(-gly).

The subclass IgG pattern seen in lung washes that were collected after the booster immunization (Figure 4) show a significantly different pattern in the animals nasally immunized with WIV with N3 and FliC (-gly) DNA that was seen in animals receiving WIV or WIV/N3 adjuvant. In the N3/FliC-DNA groups of both outbred NMRI and inbred C57BL/6 a significantly stronger influenza-specific IgG2-response was detectable in lung washes, which suggested a more balanced Th2/Th1 immunresponse against the H1 hemagglutinine antigen.

### 3.3. Hemagglutination Inhibition

Although increases in serum HAI titers were observed at day 21 when comparing WIV to WIV and adjuvant groups, significantly increased HAI titers were only detectable after booster immunizations (Figure 5).

Booster immunization of both mouse strains with WIV with N3 or N3 and pFliC(-gly) significantly raised their HAI titer against A/H1N1/SI virus between four- and 32-fold. In mice receiving WIV immunization alone or WIV with pFliC(-gly), at best a non-significant doubling of HAI titer was seen. However, when combined with N3, pFliC(-gly) was able to promote a significant increase in HAI over WIV with N3 alone. None of the tested mice from any of the groups showed HAI titers against the A/H3N2/Wuhan strain. In general, the C57BL/6 mice developed lower serum HAI titers. These results indicate that the immune responses that were elicited by WIV together combined with N3 and N3 with pFliC(-gly) adjuvants were able to elicit clear increases in HAI titer that were well above the benchmark level of ≥40. Among the NMRI mice, all of the influenza immunized groups had animals that developed HAI-serum titer of 40 or more. The most significant responses were seen in groups where N3 and N3 combined with FliC-DNA was used as adjuvants, where all animals/group developed HAI antibody titers and the highest serum titers reached 65,000 at day 90. Among C57BL/6 mice, only animals in the two groups where N3 adjuvant was used with influenza antigen responded with HAI titers over 40. The highest HAI titers were seen in the group receiving N3 and FliC-DNA as adjuvant, with the highest HAI titers of 2560 being obtained at day 90 post-immunization.

### 3.4. Interleukin-5 Release Responses

A significantly higher amount of Interleukin-5 (IL-5) secretion was produced from animals immunized with WIV combined with adjuvants when the spleen cells at day 90 were stimulated in vitro with WIV Influenza A virus (A/H1N1/SI) (Figure 6).

The highest average amounts were observed in NMRI mice (Figure 6A) in WIV with N3 adjuvant as compared to WIV alone. The addition of pFliC(-gly) to WIV led to lower IL-5 production, however this difference was not significant. However, NMRI mice receiving WIV and N3 with pFliC(-gly) had a significantly lower secretion of IL-5 production after influenza antigen restimulation. In naïve mice only given adjuvant, no IL-5 secretion was seen when stimulated with WIV. Restimulated spleen cells from C57BL/6 mice produced, on average, one-third of the IL-5 amounts that were observed in NMRI mice (Figure 6B). Cells from all mice immunized with WIV and adjuvant produced significantly higher IL-5 amounts than WIV alone immunized mice. However, no significant difference in IL-5 production was observed between the WIV and N3 vaccinated mice and those that were given WIV and pFliC(-gly) or WIV with N3 and pFliC(-gly). Together, these results demonstrate that WIV immunization with N3 leads to cellular immune responses that were capable of IL-5 production, which can be attenuated by the addition of pFliC(-gly).

### 3.5. Systemic Cell-Mediated Immunity

T cell-mediated immune responses to influenza A/H1N1/SI, NP-peptides, and A/H3N2/Wuhan virus showed significantly higher IFNγ spot reactivity in animals that were immunized with WIV with N3 and pFliC(-gly) adjuvant. Already, after one immunization (Figure 7A, day 21) IFNγ secreting spleen cells responding to A/H1N1/SI were significantly increased in NMRI and C57BL/6 mice that were immunized with WIV and N3 or WIV with N3 and pFliC(-gly) adjuvants.

After one booster immunization, IFNγ responses to A/H1N1/SI in both strains of mice were significantly increased in the WIV immunized groups if adjuvant was used (Figure 7B, day 90) as compared to non-adjuvanted immunizations.

Peptides from the conserved nucleoprotein (NP) from influenza A were selected to characterize the responses to a defined CTL-epitope. A H2d NP (aa 147-156) binding peptide was chosen to restimulate NMRI mice, and NP (aa 55-64) was chosen to restimulate H2b C57BL/6 mice. Mice in the groups given WIV with N3 responded by developing significantly higher numbers of IFNγ secreting cells than animals receiving WIV alone (Figure 7C). Adjuvant effects were further enhanced in WIV with N3 groups by the addition of pFliC(-gly).

A significant reduction in IFNγ ELIspot reactivity was only observed animals in groups where pFliC(-gly) was used as adjuvant when CD8+ T cell depletions were performed (Figure 7D). These results reveal the proportion of responses that are derived from CD8+ cells, but also indicate that all of the influenza responding mice develop IFNγ secreting CD4+ T cells against influenza. In all experiments, mice that were immunized with WIV alone developed no or very few IFNγ ELIspot secreting cells (7–55 spots/spleen million cells), which was not significantly higher than influenza naïve control mice (Figure 7A–D).

Heterosubtypic cell-mediated IFN-γ secreting immunity was tested while using A/H3N2/Wuhan influenza as a recall antigen (Figure 8).

In general, the total numbers of IFNγ secreting cells were two-fold lower than those that were elicited by the homologous influenza strain (H1N1). Among the NMRI mice, all groups of immunized with WIV and N3 adjuvant developed significantly higher numbers of IFNγ secreting cells than the mice receiving WIV alone. A similar trend was observed among the C57BL/6 mice. The addition of the pFliC(-gly) adjuvant lead to even greater numbers of IFNγ secreting cells responding to H3N2.

Together, these results demonstrate that the addition of N3 adjuvant significantly elicits splenic T cell responses to homotypic whole influenza A after just one vaccination and after a single boost enhances the responses even further. The addition of pFliC(-gly) to a WIV and N3 immunization was, in nearly all cases, able to greatly enhance the immune responses when compared to WIV and N3 alone. Analysis of T cell responses after WIV and N3 boosting also revealed an ability to respond to conserved Class I T cell epitopes, as well as heterotypic influenza A strains. The addition of pFliC(-gly) to WIV and N3 immunizations were also able to greatly enhance these responses, but the analysis of specific T cell populations additionally revealed that a significant portion of immune reactivity came from both CD8 as well as CD4-expressing cells.

Summarizing the cell-mediated immune responses that were obtained at day 90, prior to influenza virus challenge, suggest that the used adjuvants were all capable of supporting both influenza-antigen stimulated IFN-γ secreting and IL-5 secreting immunity in vitro. The highest levels of IFN-γ secreting responses were detectable in the animals (of both strains of mice) given WIV and FliC-DNA/N3 adjuvant, of which around 50% seem to be from CD8+ T cells against the tested NP-epitope in the spleen of C57BL/6 animals (Figure 7D) and with cross-reactivity towards influenza A/H3N2 virus antigen (Figure 8). Thus, the most pronounced and broad influenza-specific immune responses were obtained through the combination of WIV A/H1N1/SI with FliC (-gly) and N3 adjuvant administered twice nasally.

The potential correlates of protective immunity on long-term periods were tested after in vivo challenge with a heterologous influenza A strain (California H1N1/2009pdm strain) and systemically analyzed in spleen cells, and in mucosal samples from lung wash samples.

Nasal challenge with influenza A virus resulted in significantly better survival in mice immunized with Influenza A vaccine and adjuvants N3 and N3 + FliC(-gly) DNA. This was seen in both strains of mice (Figure 9A–D).

Interestingly, the group survival data in both in- and outbred animals fit well with the pre-immunization immune responses that were measured, where especially elevated IFN-gamma levels after influenza-peptide stimulation in vitro and subclass IgG pattern with higher influenza-antigen ELISA binding IgG2 levels, seem to be associated with increased survival. In this study, the number of survivors post challenge was prolonged with at least three months in comparison with the inbred C57BL/6 mice. N3 with FliC-DNA plasmids provided the most elevated levels of both Th1 and Th2 type immune signaling, since almost all the studied immune parameters in the study (humoral responses: influenza virus specific HAI (Table 3). lung-IgG and IgA ELISA binding antibodies, subclass IgG pattern in serum and lung wash and IFNgamma ELIspot and cytokine release pattern in vitro) show that the adjuvant combination. The data may indicate that all of these immune parameters may need to be activated in elderly animals, since, in groups immunized with single adjuvant, obtaining good Th2-type humoral immune responses, at a higher age were not as efficiently protected when challenged with pathogenic influenza virus nasally.

There is a variable test sample timepoint difference between the animals that are presented in Figure 9, Figure 10 and Figure 11. Animals from groups that more rapidly became ill after nasal influenza challenge where spleen cells were collected at day 9 to 15, at the day when they had to be sacrificed due to pathogenicity. From the groups where animals better resisted the influenza virus challenge (among NMRI mice groups 2 and 4 and C57BL/6 mice groups 7 and 9), the spleens were collected at day 29–30.

The IFN-gamma ELIspot analyses in splenocytes that were collected from mice challenged with heterologous A/H1N1/pdm09 virus without a FluA vaccination with potent Th1-type enhancing adjuvant illustrate that, if mice are allowed to reach old age, their cell-mediated immunity responds to slowly to protect from disease. Even though they have previously responded with a substantial influenza A neutralizing (HAI) response and at least a detectable influenza-antigen binding serum and mucosal IgG and IgA ELISA response, with time due, to aging on poor stimulation the levels may drop to low levels. This seems to be the case both for in- and outbred mouse strains.

Avidity Index (AI) against the recombinant HA of A/H1N1/Ca09pdm was significantly higher in the serum samples that were collected from mice immunized with N3 + FliC-DNA adjuvant (Groups 4 and 9, *p* < 0.01). Prior to challenge, the median AI in group 4 was 0.97 (0.86–1.11) and in group 9 median AI 0.86 (0.78–0.91) in comparison with the other influenza vaccinated groups with median AI 0.34 (0.09–0.46).

An attempt to perform mucosal influenza A/H1N1neutralization assays was performed against the challenge virus. However, the amount of IgA was quite low in each individual washing solution, so to perform the assay pooling and the concentration of samples was needed. Thus, we obtained one single pool from each study group for a single assay effort. The obtained results showed a HAI titer of 40, and only in the lung wash pool from the group of NMRI mice that were immunized with the N3 and FliC-DNA adjuvant.

It seems clear that the analyzed influenza-antigen stimulated cell-mediated immunity, both before challenge (Figure 7 and Figure 8) and after influenza challenge (Figure 10, Figure 11 and Figure 12), the later in old animals, the vaccine regimen containing adjuvants that enhance both humoral, virus-neutralizing, and binding antibodies in serum and respiratory mucosa, together with interferon-gamma secreted cell-mediated immunity seem to result in long-lasting protective immunity in both strains of mice, but perhaps more in out-bred than in inbred animals.

## 4. Discussion

A single-dose vaccination would be highly useful, especially in emergency situations with rapid spread of influenza, as most inactivated influenza vaccines require two immunizations to provide full and protective immunity [30,31]. Previous mouse studies with formalin-inactivated WIV have shown that the rapid development of B cell responses in serum and upper airway mucosa is sufficient to protect from mortality. We assessed the use of two potential mucosal adjuvants in combination with WIV intranasal vaccination to improve upon these results and observed the induction of mucosal B cell responses as well as heterosubtypic systemic cellular responses that were detectable after only one dose. Immune reactivity is further enhanced after a nasal booster immunization. Thus, to investigate the development of long-term immunity, vaccinated animals were studied over nine months post-booster immunization before a heterologous influenza A/H1N1 challenge experiment was performed, in mice reaching 21–23 months of age.

Nasal administration of inactivated influenza A antigen with N3 and DNA-encoded flagellin from *Salmonella typhimurium* or from flagellin isolated from human flora *Escherichia coli* significantly increased the influenza A specific immune responses. The observed immunogenicity of this low HA dose, influenza vaccine (25 μg total protein/mouse containing 1.5 μg HA) in each mouse strain induced similar levels of serum antibody, which suggested that the basic antibody responses were comparable. With addition of either N3 or N3 and pFliC(-gly) adjuvants responses were significantly enhanced in both NMRI and C57BL/6 mice after a single immunization. However, the ability of N3 and pFliC(-gly) to promote enhanced serum responses over that of N3 alone was not observed.

After the booster immunization, significantly higher serum IgG or IgA titer toward influenza antigens was only seen in the mice receiving N3 adjuvant alone or N3 and pFliC(-gly). An analysis of serum IgG isotypes revealed that the inclusion of pFliC(-gly) in the adjuvant composition enhanced IgG2a responses when compared to the use of N3 alone, demonstrating its ability to promote a Th1-type or a mixed Th1/Th2-type immune response. Apart from being an indicator of the degree of cell-mediated cytotoxic immune responses, the IgG2 antibodies have also been described to be more beneficial antiviral isotype antibodies. They may enhance antigen uptake and cross-presentation due to their higher capacity to bind to Fc-receptors on antigen-presenting cells, as well as demonstrating better C’-activating properties via the C1q-pathway [32]. The capacity of non-coding plasmid DNA formulated with vaccine antigens to improve or strengthen Th1- or mixed Th1/Th2-type antibody and cell mediated immune responses has been documented [33,34]. However, when compared to our previous study that demonstrated a specific adjuvant effect by the FliC open-reading frame [25], our results suggest that FliC is exerting similar immune enhancing responses observed here.

It is not a straightforward task to understand how such a vaccine formulation functions due to the complex nature of our antigen adjuvant formulation(s). However, we favor a model where the nasal delivery of a cationic adjuvant N3 affects the mucus at mucosal surface layers to better penetrate and deliver the influenza vaccine, together with pFliC(-gly). This would allow for larger amounts of WIV antigen to be taken up by antigen-presenting cells, leading to antigen processing as well as the potential for viral ssRNA to stimulate TLR7 [35]. This enhancement may account for the adjuvant effect of N3 alone. When combined with pFliC(-gly), the N3 may allow for pFliC(-gly) “transfection” of cells in the mucosa triggering multiple innate immune receptors by way of activating dsDNA sensors, as well as sequence-specific CpG DNA sensors [35]. Additionally, as a consequence of FliC polypeptide production, NLRC4-triggered inflammasome activation and inflammatory cell death could occur in the permissive cells [36]. The secretion of flagellin from living or release from dying cells may also activate TLR5 [36,37]. Irregardless of the mechanism, we clearly observe that adjuvant the inclusion of pFliC(-gly) stimulates a mixed Th1/Th2-immune response pattern in both inbred and outbred mice, indicating its benefit in broadening immune responses to WIV.

The use of adjuvants in influenza vaccinations may be especially valuable in elderly individuals that otherwise fail to respond or lack strong influenza virus-specific cell-mediated immunity after traditional adjuvant free vaccination. However, one need to bear in mind is that influenza vaccines with adjuvants have also previously resulted in significant side effects, such as the Berna vaccine incidents where bacterial toxoid subunits, with the ganglioside GM1-binding properties were used as adjuvants for nasal immunization with the risk of resulting in Bells palsy [38,39]. The most recent serious side-effects with the intramuscularly administered influenza A/H1N1/pdm09 vaccine being given with the ASO3-adjuvant, resulting in hundreds of children responding with narcolepsy as a side-effect [40]. Obviously, every new adjuvant combined with influenza-vaccine will need to be analyzed for these undesirable side effects. One should bear in mind that this study has the main goal of providing long-lasting immunity that persists in protective immune responses at higher ages, instead of in children. The influenza strain that was used in the primary immunization belongs to the seasonal influenza strains, but the data obtained show that these influenza strains can provide protection against pandemic influenza A strains, such as A/H1N1/pdm09. Thus, the suggestion in this work is to immunize adults, to develop a robust enough immunity with the capacity to remain protective at higher ages, as exemplified in this report.

The humoral mucosal immunity that was observed in the respiratory tract of immunized mice was clearly enhanced by the presence of cationic N3 adjuvant in the inactivated influenza vaccine mixture. However, the highest lung wash IgA and IgG titers against influenza was seen in the mice of both mouse strains after immunization with N3 and pFliC(-gly). This combination of adjuvant N3 with pFliC(-gly) significantly enhanced the systemic and the mucosal immune responses (41). However, slightly different immune reaction patterns were observed between the two strains of mice when the samples were analyzed by HAI. The putatively protective and neutralizing serum HAI titer among the outbred NMRI mice initially (Day 21) shows equivalent titers for the group vaccinated with WIV alone and groups vaccinated with WIV and adjuvant (Figure 4A). After the booster immunization, the HAI-titers increased to significantly higher titers in N3, and N3 and pFliC(-gly) combination immunized groups (day 90). C57BL/6 mice more robustly responded to WIV with adjuvant after one immunization, which was likely because of strain-specific differences. Although these results indicate that immunization in outbred populations may require two doses, importantly they indicate that these adjuvants have the potential to function effectively in outbred veterinary animal populations as well as humans to promote WIV mucosal immune responses. The inclusion of these adjuvants in an intra-nasal non-living vaccine could provide enhanced protection of the upper mucosa. The most common route of influenza A transmission [41].

In vitro analysis of IL-5 secretion by splenocytes that were stimulated with WIV H1N1 from WIV and N3 immunized mice showed that cells from NMRI mice secreted significantly higher amounts of IL-5 than C57BL/6 mice. This interleukin is normally considered to be a Th2-type, which suggests that the NMRI mouse strain may be more Th2-skewed than C57BL/6. Direct comparison of our trends of IL-5 secretion to the serum IgG and IgA titers and IgG isotypes also established that higher IL-5 levels correlate with Th2-type immunity. Importantly, our observed IL-5 levels peaked at different time points in the two mouse strains. The NMRI spleen cell secretion of IL-5 peaked at 44–48 h while cells from C57BL/6 mice peaked at 70–72 h after stimulation. These differences could be due to natural differences in the immune kinetics in each strain, differences in the total responding cell populations, or fewer spleen cells being capable of IL-5 secretion in the C57BL/6 mice. The IL-5 response following mucosal vaccination illustrates its correlation to immunological aging that could not be seen in younger mice in previous reports looking at the effects of aging and cytokine responses [42] (McDonald JU et al. 2017). Despite these strain-specific differences in their ability to produce IL-5 in response to immunization with WIV and N3, this trend could be altered by the addition of pFliC(-gly) adjuvant, demonstrating its ability to skew the immune responses towards a broader and possibly more robust anti-influenza A immunity.

Preclinical influenza vaccine studies use the influenza virus HAI as a gold standard test to identify the immunization regimes that reveal most favourable vaccine candidates. Although this is not a true influenza A neutralization assay, it correlates well with clinical antibody mediated protection from infection and mortality in man and animals. Ideally, the best measure would be to perform a challenge study with the live influenza virus on influenza vaccinated ferret or man to investigate if protection can be obtained. In our study, we did not have access to a mouse-attenuated pathogenic influenza A/H1N1/SI strain, which is why homologous challenge studies were not performed. Instead, a mouse pathogenic heterologous influenza A/H1N1/pdm09 was performed at days 180 and 270 post final immunization. In lieu of this, HAI analyses with the Salomon Island strain was performed with serum from the various influenza vaccinated groups of mice. Sera with the highest HAI titers against the homologous H1N1 influenza strain were also subjected to an HAI assay against A/H3N2/Wuhan, but they were negative. Putatively protective (≥40) average serum HAI titers to the influenza vaccine strain was already seen after one immunization in the groups that received WIV with N3 or N3 and pFliC(-gly), suggesting a more rapid B cell response toward the HA antigen. These HAI titers were further elevated after the booster immunization and after boost also present in the WIV vaccinated mice given N3 together with pFliC(-gly). Interestingly, the serum and lung wash antibody ELISA titers are shown to increase, even when the HAI titers do not change or increase in a similar manner. Thus, it is important to remember that the binding antibody titers, as measured with ELISA, do not provide any information regarding how their functional activities against influenza A virus or influenza-infected cells. Instead, they may function as surrogate biomarkers of immunogenicity or (at least in mice) suggest Th1/Th2-type immune patterns against the studied antigens.

The choice of vaccination route is important to induce an effective immune response at a specific site. The inactivated vaccines that are currently used are usually intramuscularly injected, but it would be beneficial to administer the vaccines without needles. It would entail easier and faster administration, less expense, and discomfort with needles would be avoided. However, the alternative administration methods have limitations and they do not always lead to an adequate antibody response [31]. The injectable influenza vaccines are protective, because of their ability to induce influenza neutralizing serum antibodies as well as ADCC active antibodies, mainly IgG [33]. They can prevent the pathogen from spreading and be protective at the mucosal surfaces of the lower respiratory airways. However, resistance to influenza infection is connected to systemic and mucosal immunity. Serum IgA antibodies are produced to HA and NA in the upper respiratory tract, and IgG antibodies are also protective in the lower respiratory tract. In comparison to an injectable vaccine, the amount of local IgA antibodies in nasal washings is higher after nasal vaccination, but the amount of serum antibody titers is lower [36,37,41].

Our observations indicate that intra-nasal immunization with a formulation of WIV and N3 lipid alone is able to greatly enhance HAI-specific IgG and IgA in the upper respiratory tract, demonstrating that this approach can elicit effective antibodies at the location where influenza A infects. This effect can be enhanced with the addition of pFliC(-gly). Importantly, pFliC(-gly) strongly promotes a skewing of IgG isotypes towards IgG2a/c, as well as an enhancement in homotypic and heterotypic cellular immune responses. A subclass IgG1/IgG2 pattern mirroring what was seen in serum was seen also in the lung wash IgG content. However, it is unclear how well this indicates the local cell-mediated immunity in the lung tissues. Cellular immunity to influenza A has been long speculated to provide a role in limiting the pathology of influenza A infection, and recent evidence indicates that this is indeed the case [16]. Although it remains to be seen whether the promising anti-viral immune responses that we observe in mice can also be induced in humans, the approach that is delineated here indicates that it is possible to develop a non-living influenza A intranasal vaccine that is capable of eliciting both mucosal humoral antibody, as well as strong systemic cellular responses.

We kept groups of mice during a six and nine months follow up time to address the issue of long-term immunity as well as the protective capacity from disease and illness, and then nasally challenged them with a heterologous influenza A/H1N1 virus. At nine months, especially among the out bred NMRI mice, a complete protection (100%) was obtained, while the inbred C57Bl6 mice obtained 75% protective immunity, long-term, if both cationic N3 and FliC-DNA was used as adjuvant. The immune parameters that were seen in the best protected animals were cell-mediated IFN-gamma responses against a broader repertoire of influenza antigens, than in the other groups. Furthermore, mucosal IgA and serum IgG with a broad influenza A hemagglutinin antigen binding capacity was only seen in these groups and significantly more among the NMRI mice than in C57Bl6 mice, post-challenge. The fact that we used inactivated full virion antigen may explain this phenomena, as, in other studies where the FluMist vaccine was tested together with ritatolimoid (a TLR3-agonist), a broadened secretory IgA response was obtained [43]. Otherwise, no inactivated whole virion influenza vaccines are available on the vaccine markets. Thus, it is possible that the combined Th2/Th1 balanced immunity that was developed in the WIV N3+FliC groups provided a lasting protective immune mixture containing not only a cell-mediated IFN-gamma immunity against conserved NP-epitopes and mucosal broadly HA-binding IgA, but also serum IgG with influenza-specific ADCC-activity, often requiring antibodies with high binding affinity. ADCC have been described as an important immune parameter in protecting mice from severe influenza infection under experimental conditions [44]. In conclusion, the results suggest that a seasonal influenza A/H1N1/Salomon Island/2006, whole inactivated virion combined with cationic oil-in-water adjuvant with DNA-expressed Flagellin C *S. Typhimurium* given twice nasally provide protective immunity in 22 months old in- and out-bred mice against heterologous influenza A/H1N1/California/2009 challenge over nine months. However, with the recent experiences with side-effects when new adjuvants were combined with parenteral influenza vaccination, a thorough safety monitoring will be essential before aiming towards large-scale vaccination campaigns.

## Figures and Tables

**Figure 1 vaccines-07-00064-f001:**
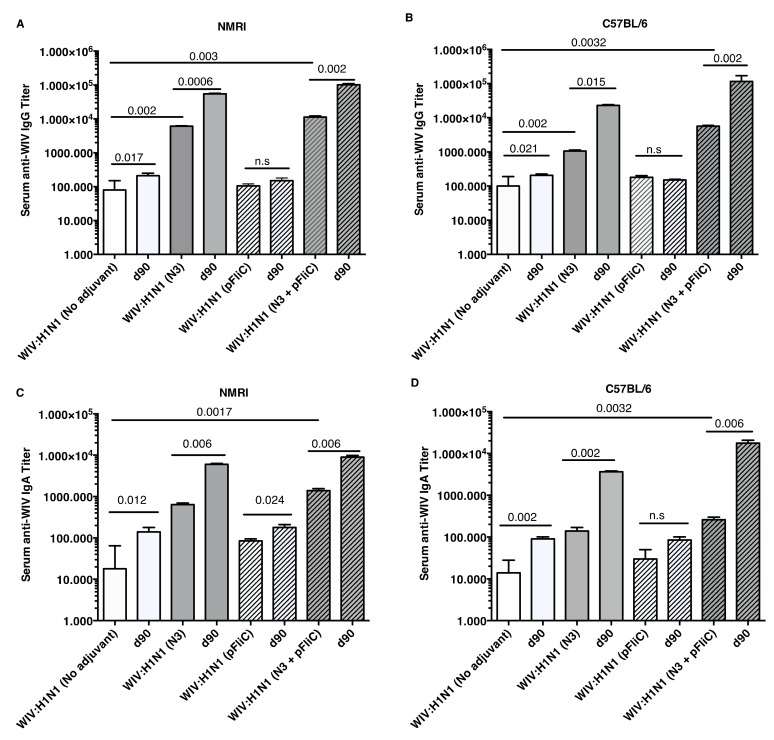
Influenza A H1N1/WIV/SI specific serum IgG and IgA titers. NMRI mice (**A**) and C57BL/6 mice (**B**) IgG titers 3 weeks (day 21) after one immunization and 4 weeks (day 90) after the booster immunization shown for each group. Each bar shows geometric mean (GMT) serum titer, and error-bars show 95% confidence interval values for each study group. Influenza A H1N1 specific serum IgA titers in NMRI mice (**C**) and C57BL/6 mice (**D**) three weeks (day 21) after one immunization and four weeks (day 90) after the booster immunization shown for each group. Bars show geometric mean (GMT) serum titer, and error-bars show 95% confidence interval values for each study group. Significant differences are indicated by nonparametric Mann–Whitney U analysis p-values. n.s. = not significant.

**Figure 2 vaccines-07-00064-f002:**
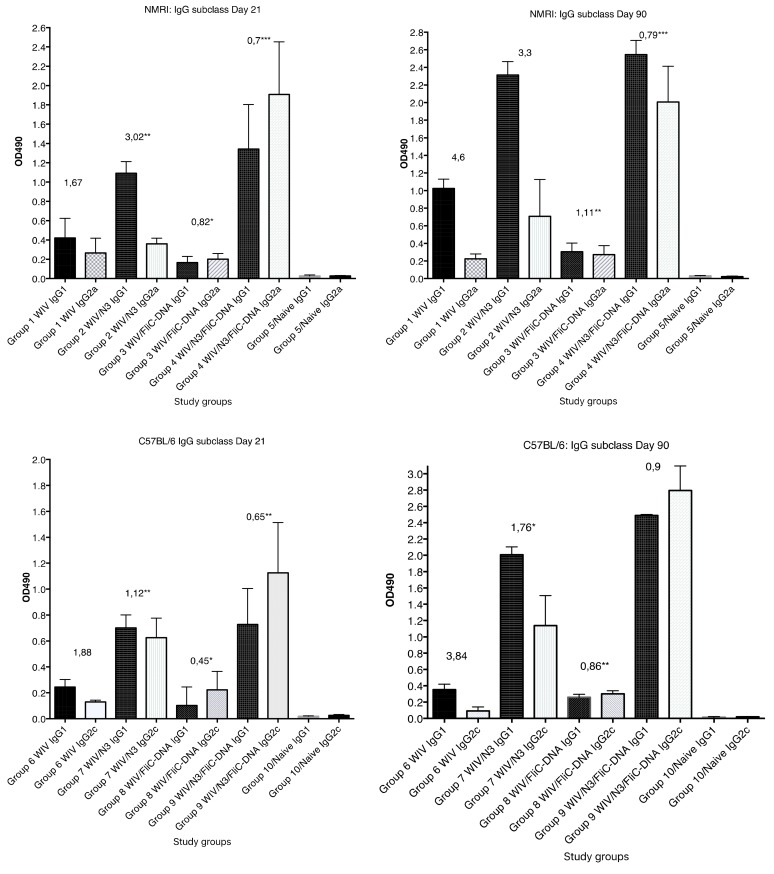
The anti-influenza A/H1N1/SI whole inactivated viral (WIV) specific serum IgG1 and IgG2a/2c subclass reactivity OD490 reactivity in serum collected at 21 days post primary immunization and 30 days post booster immunization (day 90). The median and range OD490 reactivity is shown for each study group (n = 6–8 animals/group), illustrating the results that are shown in Table 2. Significance. * = *p* ≤ 0.05, ** = *p* ≤ 0.01, *** = *p* ≤ 0.001.

**Figure 3 vaccines-07-00064-f003:**
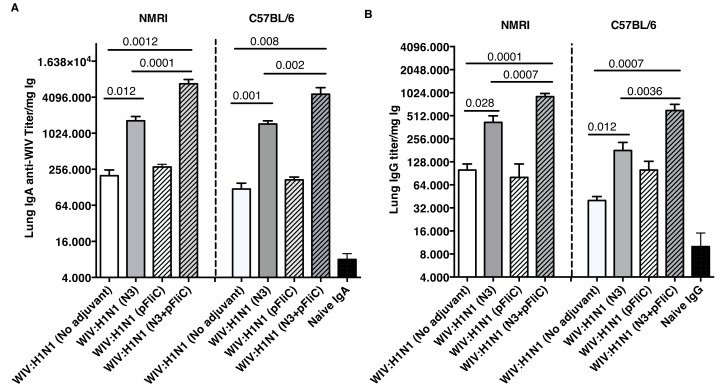
Influenza A H1N1 specific lung wash IgA and IgG. IgA anti-WIV titer in NMRI and C57BL/6 mice four weeks after booster immunization (day 90) (**A**). IgG anti-WIV titer in NMRI and C57BL/6 mice four weeks after booster immunization (day 90) (**B**). Titers presented are as anti-WIV titer/mg total IgA or IgG. Bars show GMT, and error-bars show 95% confidence intervals for each study group. Significant differences are indicated by Mann–Whitney U analysis p-values.

**Figure 4 vaccines-07-00064-f004:**
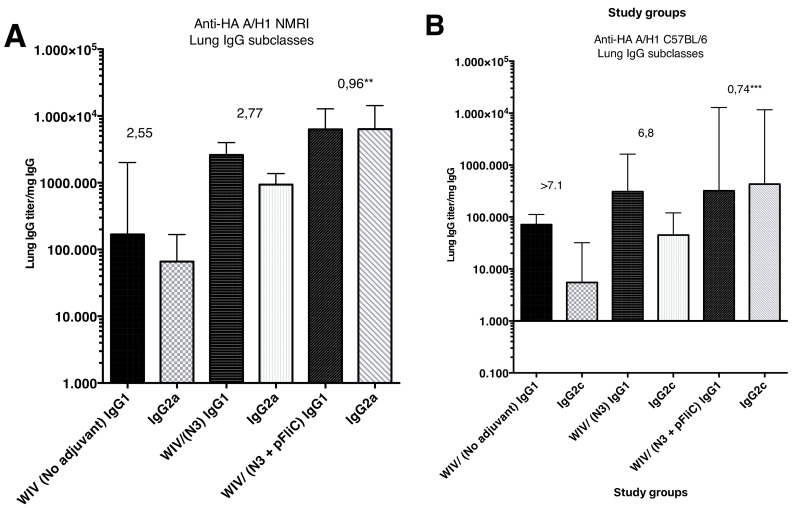
Anti-influenza A/H1N1/2009pdm rHA in lung-wash subclass IgG1 and IgG2a/IgG2c ELISA reactivity seen in three study groups at day 90 post primary immunization. (**A**) show IgG1 and IgG2a median subclass ELISA reactivity (and range) in lungs wash samples collected from outbread NMRI mice. The value given on top of each pair of bars indicates the median IgG1/IgG2a ratio in the group. (**B**) show IgG1 and IgG2c median subclass ELISA reactivity (and range) in lungs wash samples collected from inbread C57BL/6 mice. The value given on top of each pair of bars indicates the median IgG1/IgG2c ratio in the group. * = *p* ≤ 0.05, ** = *p* ≤ 0.01, *** = *p* ≤ 0.001.

**Figure 5 vaccines-07-00064-f005:**
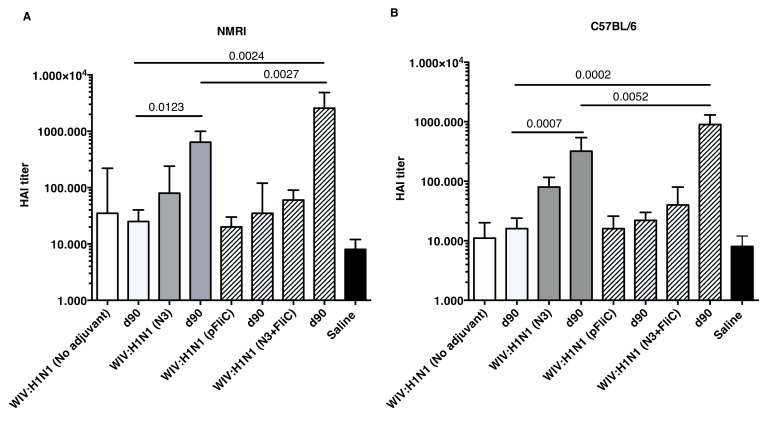
Influenza A H1N1/SI specific serum HAI titers. Titers in NMRI mice (**A**) three weeks after one immunization (day 21) and 4 weeks after the booster immunization (day 90) for each group of mice. Titers in C57BL/6 mice (**B**) three weeks after one immunization and four weeks after the booster immunization for each group of mice. Bars show GMT serum titer, and error-bars show 95% confidence intervals for each study group. Significant differences are indicated by nonparametric Mann–Whitney U analysis *p*-values.

**Figure 6 vaccines-07-00064-f006:**
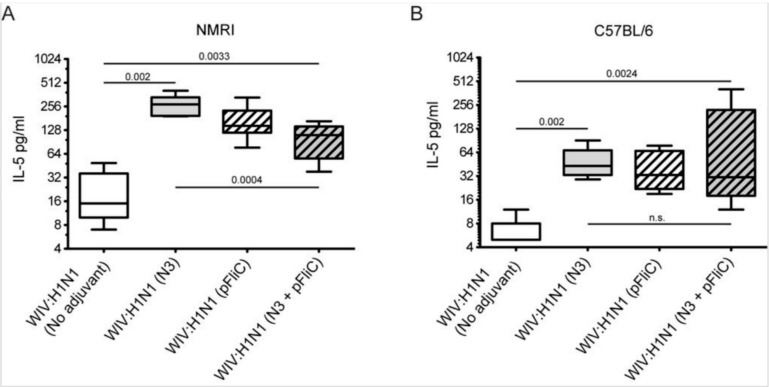
Splenocyte IL-5 release in response to Influenza A recall. (**A**) Influenza A specific IL-5 secretion (median pg/mL) million spleen cells at 48 h in NMRI mice, four weeks after booster immunization. (**B**) Influenza A specific IL-5 secretion (median pg/mL) million spleen cells at 72 h in C57BL/6 mice, four weeks after booster immunization. Block figures show median culture medium IL-5 concentrations after in vitro stimulation with WIV A/H1N1/SI, and error-bars show maximum and minimum values for each study group. The different time points chosen for the two mouse strains were determined in an in vitro pre-study influenza/ConA stimulation of spleen cells, and the optimal time point for highest levels of IL-5 secretion in spleen cells of adjuvant-vaccinated aged animals was chosen. Significant differences are indicated by nonparametric Mann–Whitney U test *p*-values.

**Figure 7 vaccines-07-00064-f007:**
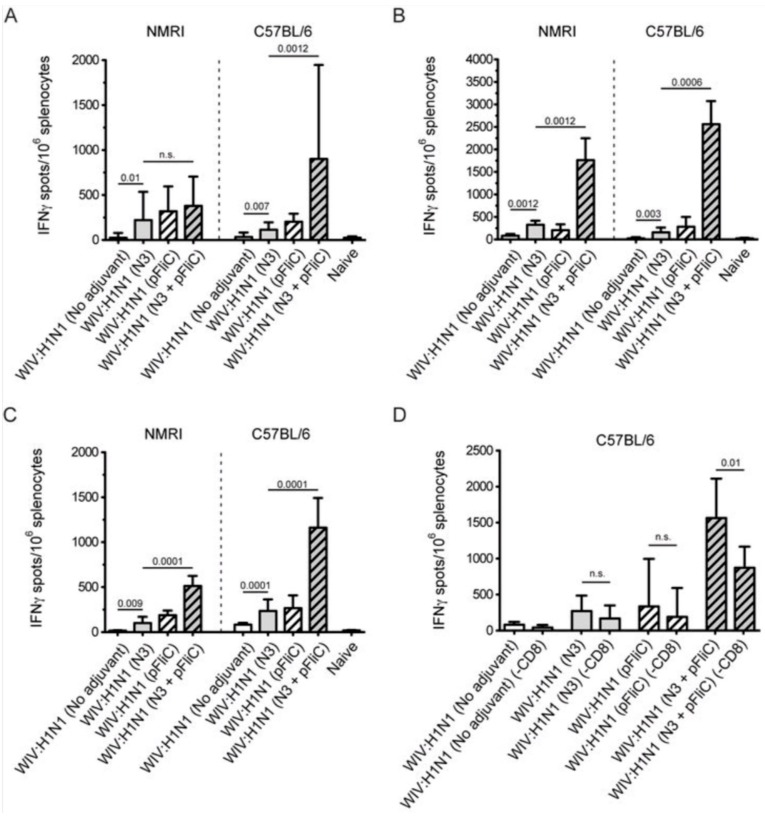
Splenocyte anti-Interferon-γ (IFNγ) producing cell frequency (ELIspot) in response to Influenza A antigen recall. (**A**) Influenza A/H1N1/SI specific IFNγ producing spleen cells in NMRI mice (left side) and C57BL/6 mice (right side), three weeks after primary immunization. (**B**) Influenza A/H1N1/SI specific IFNγ producing spleen cells in NMRI mice (left side) and C57BL/6 mice (right side), four weeks after booster immunization. (**C**) Influenza A/NP-peptide specific IFNγ producing spleen cells in NMRI mice (left side) (against peptide; NP147-156/aa TYQRTRALV) and C57BL/6 mice (right side) (against peptide, NP 366-374 aa ASNENMDAM), four weeks after booster immunization. (**D**) Influenza A/H1N1/SI specific IFNγ producing CD8-depleted spleen cells in C57BL/6 mice, four weeks after booster immunization. Bar-height shows geometric mean (GMT) spots/million cells, and error-bars show 95% confidence intervals for each study group. Significant differences are indicated by nonparametric Mann-Whitney U analysis p-values.

**Figure 8 vaccines-07-00064-f008:**
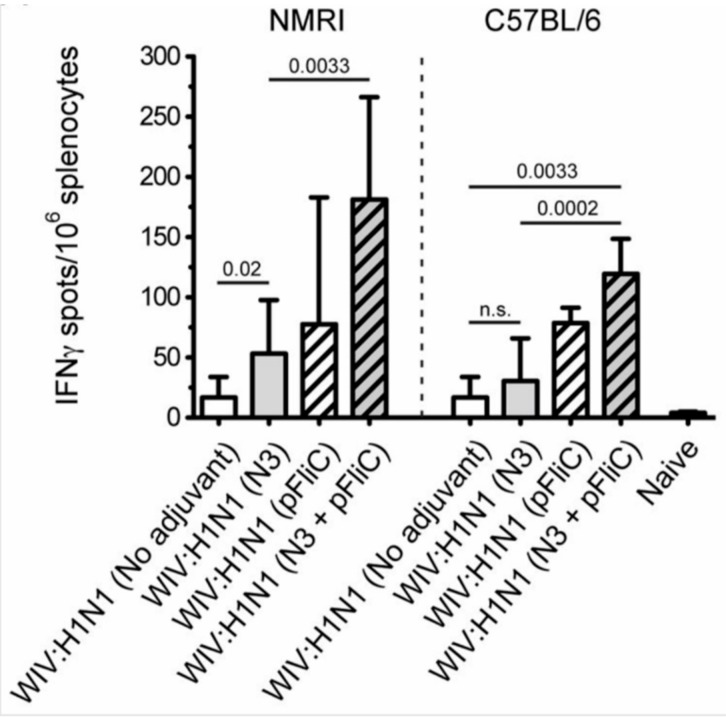
Splenocyte IFNγ producing cell frequency (ELIspot) in response to influenza A/H3N2/Wuhan stimulation. Influenza A/H3N2/Wuhan specific IFNγ spleen cells in NMRI mice (left side) and C57BL/6 mice (right side), 4 weeks after booster immunization. Bar-height shows geometric mean (GMT) spots/million cells, and error-bars show 95% confidence intervals for each study group. Significant differences are indicated by nonparametric Mann–Whitney U analysis *p*-values.

**Figure 9 vaccines-07-00064-f009:**
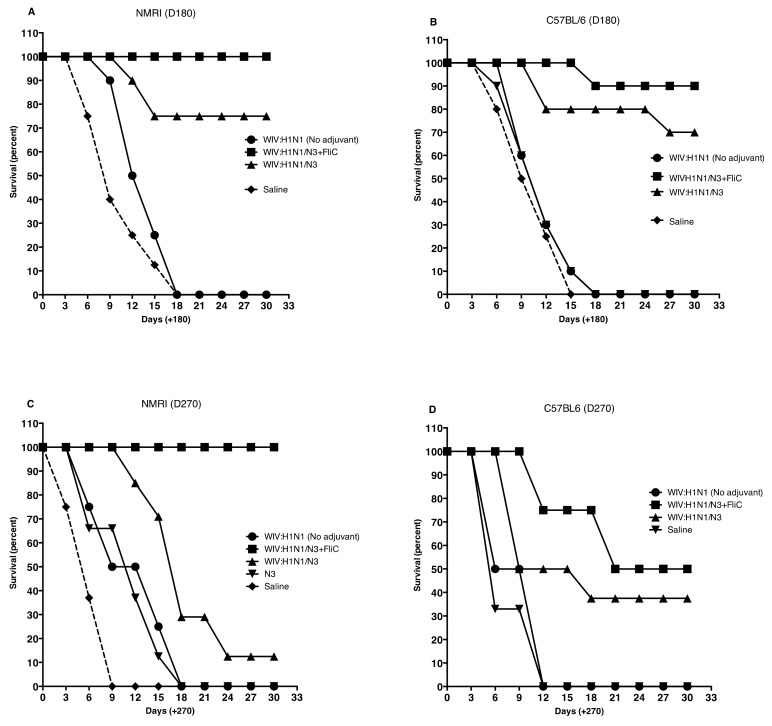
Kaplan-Meier graphs from day 180 post challenge show significantly better survival (100% to 87.5%) in NMRI mice (**A**) and in C57Bl/6 mice (**B**) if the WIV influenza vaccine was combined with N3 and FliC-DNA. as adjuvant and 80–75% survival when WIV with N3 adjuvant was used. (**C**,**D**). Kaplan-Meier graphs from day 270 post-challenge show significantly better survival (100% to 50%) in NMRI mice (**C**) and in C57Bl/6 mice (**D**) if the WIV influenza vaccine was combined with N3 and FliC-DNA. as adjuvant and 20–37.5% survival when WIV with N3 adjuvant was used.

**Figure 10 vaccines-07-00064-f010:**
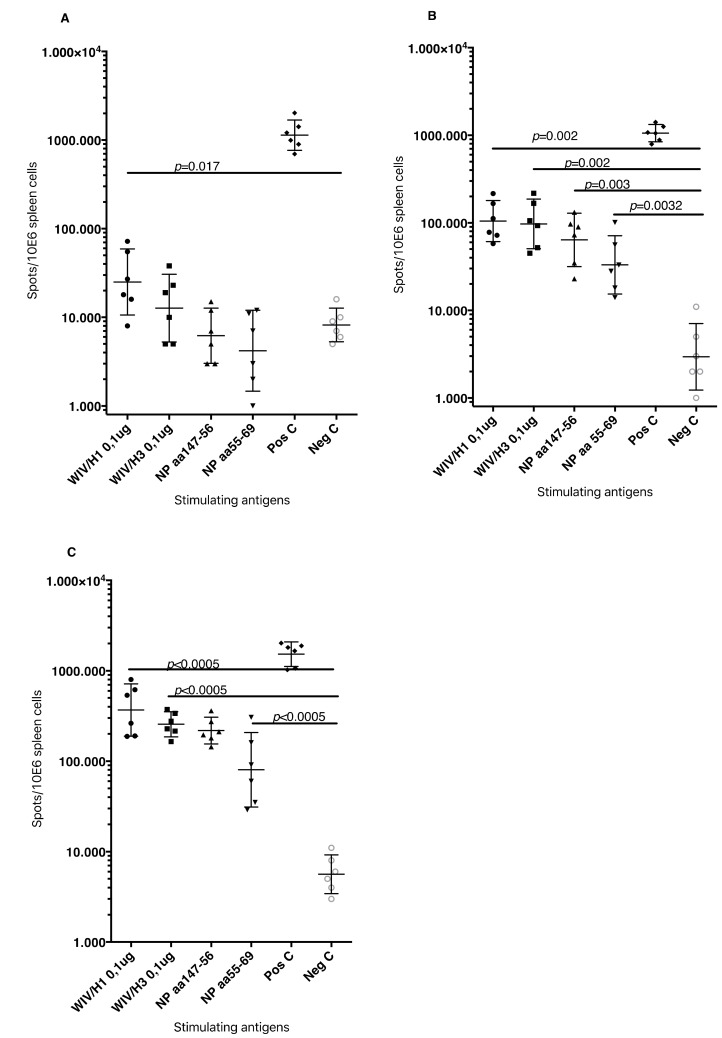
Cell-mediated immunity against influenza antigens after A/H1N1/California.pdm09 challenge was shown as IFN-gamma responses in stimulated spleen cells in vitro in three study groups of vaccinated NMRI mice (Figure 10A–C). (**A**) illustrates the GMT (95% C.I) IFN-gamma ELIspot responses in WIV vaccinated mice (no adjuvant) at day the day of sacrifice day 9–18, (**B**) the WIV with N3 as adjuvant, at the day of sacrifice, at days 12–30, and (**C**) the WIV with N3 and FliC-DNA as adjuvant, at day of sacrifice, at day 29–30. The frequencies of spots/million were evaluated against WIV/Influenza A/H1 and A/H3, as well as against two CTL-peptides from the NP-protein. Significant differences are indicated by nonparametric Mann–Whitney U analysis *p*-values.

**Figure 11 vaccines-07-00064-f011:**
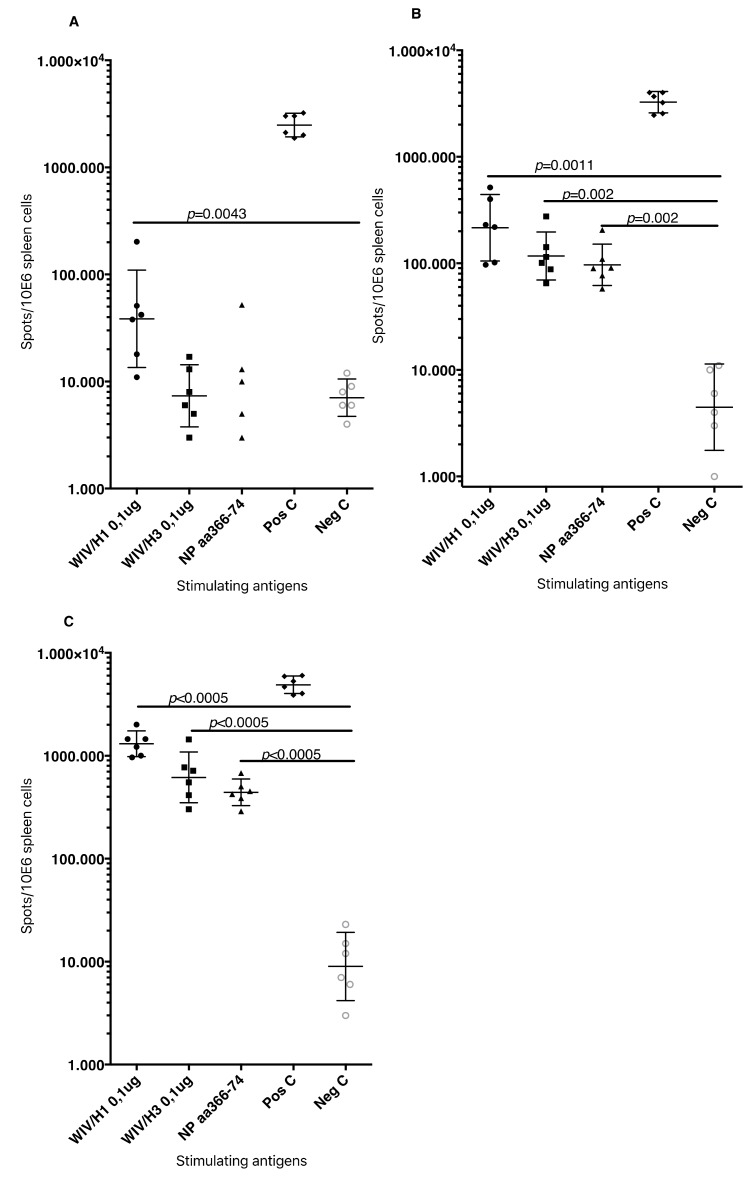
Cell-mediated immunity against influenza antigens after challenge was shown as IFN-gamma responses in stimulated spleen cells in vitro in three study groups of vaccinated C57BL6 mice (**A**–**C**). (**A**) illustrates the GMT (95% C.I) IFN-gamma ELIspot responses in WIV vaccinated mice (no adjuvant) at day of sacrifice at days 6, 9, and 12, (**B**) the WIV with N3 as adjuvant, at day of sacrifice at days 9, 18, and 30 and (**C**) the WIV with N3 and FliC-DNA as adjuvant, at day of sacrifice at days 12, 21, and 30. Frequencies of spots/million were evaluated against WIV/Influenza A/H1 and A/H3, as well as against one CTL-peptide from the NP-protein. Significant differences are indicated by nonparametric Mann–Whitney U analysis *p*-values.

**Figure 12 vaccines-07-00064-f012:**
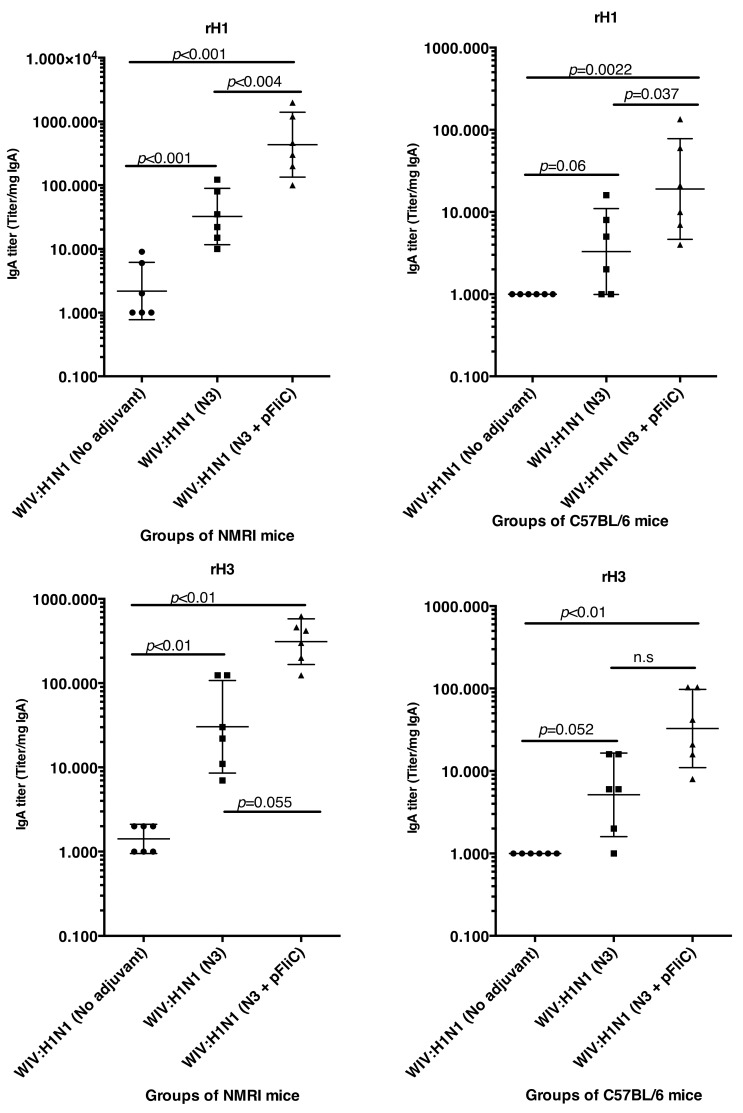
Mucosal lung IgA in lung wash fluids from NMRI and C57BL6 mice, post challenge. IgA anti-recombinant HA/H1/CA09 antigen (**A**,**B**) and anti-recombinant HA/H3/CA (**C**,**D**) ELISA titer analysis. Significant differences are indicated by nonparametric Mann–Whitney U analysis *p*-values.

**Table 1 vaccines-07-00064-t001:** Study design and nasal immunization of NMRI and C57BL/6 mice against formalin-inactivated Influenza A/H1N1/Salomon Island/2006.

	Immunization Schedule				
Group	n	Immunization Days	Dose and Immunogen	Adjuvant	Mouse Strain
1	32	0, 60	1.5 µg HA in whole intact inactivated H1N1	None	NMRI
2	32	0, 60	1.5 µg HA in whole intact inactivated H1N1	N3	NMRI
3	16	0, 60	1.5 µg HA in whole intact inactivated H1N1	pFliC(-gly)	NMRI
4	32	0, 60	1.5 µg HA in whole intact inactivated H1N1	N3 + pFliC(-gly)	NMRI
5	30	0, 60	Saline	N3 + pFliC(-gly)	NMRI
6	32	0, 56	1.5 µg HA in whole intact inactivated H1N1	None	C57BL/6
7	32	0, 56	1.5 µg HA in whole intact inactivated H1N1	N3	C57BL/6
8	12	0, 56	1.5 µg HA in whole intact inactivated H1N1	pFliC(-gly)	C57BL/6
9	32	0, 56	1.5 µg HA in whole intact inactivated H1N1	N3 + pFliC(-gly)	C57BL/6
10	24	0, 56	Saline	N3 + pFliC(-gly)	C57BL/6

Abbreviations: HA = Hemagglutinine protein from influenza A, pFliC(-gly) = Plasmid encoding secreted Flagellin type C of *Salmonella typhimurium* with mammalian glycosylation signal sequences removed.

**Table 2 vaccines-07-00064-t002:** Humoral IgG subclass IgG1/IgG2a or IgG2c immune responses to Influenza A after nasal immunization of NMRI and C57BL/6 mice.

	Immunization Schedule		Humoral Anti-Influenza Specific Immune Responses					
	Dose and Immunogen	Adjuvant	Day 21		Subclass IgG 1/2a Ratio	Day 90		Subclass IgG 1/2a Ratio
			IgG1	IgG2a	Day 21	IgG1	IgG2a	Day 90
1	1.5 µg HA in whole intact inactivated H1N1	None	0.421 (0.179–0.624)	0.266 (0.094–0.419)	1.67	1.025 (0.306–1.131)	0.225 (0.099–0.279)	4.6
2	1.5 µg HA in whole intact inactivated H1N1	N3	1.091 (0.883–1.212)	0.360 (0.198–0.444)	3.02 **	2.313 (1.818–2.466)	0.707 (0.611–1.127)	3.3
3	1.5 µg HA in whole intact inactivated H1N1	pFliC(-gly)	0.231 (0.089–0.166)	0.259 (0.141–0.202)	0.82 *	0.305 (0.239–0.404)	0.273 (0.179–0.375)	1.11 **
4	1.5 µg HA in whole intact inactivated H1N1	N3 + pFliC(-gly)	1.343 (0.863–1.803)	1.909 (1.725–2.454)	0.7 ***	2.544 (2.161–2.707)	2.007 (1.933–2.414)	0.79 ***
5	None	N3 + pFliC(-gly)	0.029 (0.016–0.035)	0.026 (0.022–0.033)	n.a.	0.031 (0.019–0.035)	0.022 (0.017–0.028)	n.a.
					Subclass IgG1/2c Ratio			Subclass IgG1/2c Ratio
			IgG1	IgG2c	Day 21	IgG1	IgG2c	Day 90
6	1.5 µg HA in whole intact inactivated H1N1	None	0.244 (0.202–0.302)	0.130 (0.079–0.143)	1.88	0.353 (0.277–0.421)	0.092 (0.056–0.141)	3.84
7	1.5 µg HA in whole intact inactivated H1N1	N3	0.701 (0.599–0.801)	0.625 (0.406–0.777)	1.12 **	2.007 (1.799–2.105)	1.138 (0.455–1.506)	1.76 *
8	1.5 µg HA in whole intact inactivated H1N1	pFliC(-gly)	0.102 (0.088–0.245)	0.223 (0.206–0.365)	0.45 *	0.259 (0.191–0.295)	0.301 (0.280–0.338)	0.86 **
9	1.5 µg HA in whole intact inactivated H1N1	N3 + pFliC(-gly)	0.728 (0.524–1.005)	1.126 (1.021–1.514)	0.65 **	2.489 (2.290–2.501)	2.796 (2.611–3.096)	0.9
10	Saline	N3 + pFliC(-gly)	0.019 (0.016–0.022)	0.015 (0.011–0.020)	n.a.	0.017 (0.015–0.021)	0.019 (0.015–0.022)	n.a.

Abbreviations. * = *p* ≤ 0.05, ** = *p* ≤ 0.01, *** = *p* ≤ 0.001, n.a = Not analysed. Median OD490 nm values are shown for each group of mice (n = 6–8 mice/test day and group). Values in parenthesis show OD range.

**Table 3 vaccines-07-00064-t003:** Serum antibody reactivity in blood samples collected at day of sacrifice in influenza A/H1N1/California/2009 challenged mice. Serum was tested for their hemagglutination inhibition assay (HAI) titer against the challenge influenza and by IgG ELISA titration against the challenge virus in samples collected two weeks prior to challenge and in serum collected at day of sacrifice post-challenge.

Serum Reactivity Against Influenza A/H1N1/Ca09pdm GMT and (Range) Pre- and Post-Challenge						
			HAI		ELISA Titers	(rHA/H1N1/09pdm)
Group	Antigen	Adjuvant	Titer Pre-	Titer Post-	Pre-Chall. IgG Titer	Post-Chall. IgG Titer
			Challenge	Challenge		
**NMRI**						
1	1.5 µg HA	No	<10	<10	140 (<50–180)	200 (60–240)
2	1.5 µg HA	N3	<10	10 (<10–30)	2220	5880
					(1600–4850)	(3800–11240)
4	1.5 µg HA	N3 + FliC	<10	30 (20–60)	26770	106800
					(13,800–38,550)	(46,560–224,450)
5	Saline	No	<10	<10	<100	<100
**C57BL/6**						
6	1.5 µg HA	No	<10	<10	<50 (<50–80)	75 (50–90)
7	1.5 µg HA	N3	<10	<10	1820	4240
					(480–3130)	(3330–1550)
9	1.5 µg HA	N3 + FliC	<10	20 (<10–20)	5980	38820
					(5000–8690)	(24,450–88,580)
10	Saline	No	<10	<10	<100	<100

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
