# Peer review of "Long-Lasting Mucosal and Systemic Immunity against Influenza A Virus Is Significantly Prolonged and Protective by Nasal Whole Influenza Immunization with Mucosal Adjuvant N3 and DNA-Plasmid Expressing Flagellin in Aging In- and Outbred Mice"

_vaccines, 2019, doi:10.3390/vaccines7030064_

Reviewer 1 Report

This is a very comprehensive work with a lot variables and mouse groups. The data clearly shows that in mice nasally administered inactivated influenza A virus vaccine induce very high antibody levels and cell mediated immunity when coupled to N3 adjuvant and flagellin from Gram-negative bacteria. The work has been conducted with care, mouse groups are sufficiently high as well as a lot of immunological parameters have ben analysed and presented. As whole the paper is very long and could be condensed by putting some of the tables (e.g. table 1 and 2) and figures (e.g. fig. 6 and 10) into supplementary files. This would likely enhance the readability of the paper. There are some specific comments that the authors may find usefull in further processing of the manuscipt.

The authors did not describe any side effects that could have been caused by various vaccine compositions. Especially the addition of N3 adjuvant and flagellin is highly immunostimulatory, which could cause some nonspecific strong inflammatory responses within few days after vaccination. Was such responses seen?

The paper should shortened in order to make it more easily readble.

Some linguistic errors should be corrected.

Author Response

We have now done our best to respond on the comments and questions provided by the reviewers of our manuscript entiteled: “Long-Lasting Mucosal and Systemic Immunity Against Influenza a Virus Is Significantly Prolonged and Protective by Nasal Whole Influenza Immunization with Mucosal Adjuvant N3 and DNA-Plasmid Expressing Flagellin in Aging In- and Outbred Mice.

We here below provide our responses, point-by-point and enclose the now re-written and language corrected manuscript. Our changes in the text are indicated with red text.

Reply on Reviewer 1:s questions and comments.

We thank reviewer 1 for valuable and insightful comments.

Q. We thank the reviewer 1 for the suggested idea that the manuscript could include a discussion regarding adjuvants and side effects.

Answer: We have now included text concerning these issues, where we exemplify how addition of adjuvants to influenza vaccines can result in severe side effects. Our comments are added in the Discussion section.

In the mice that received the used adjuvants, in these low concentrations did not show any signs of unwanted effects apart from some sneezing and dizziness, most likely from the Isofluran sedation during the intranasal administrations.

Q. Reviewer 1 also suggest that some of the tables and figures could be moved to the Supplementary section to enhance the readability of the manuscript.

Answer: We were unaware of this possibility of using a Supplementary section. We will be happy to move the suggested data into this section.

Q. The English language need to be checked.

Answer: Yes, we have sent the re-written amuscript for English language edition, and we hope this new version will now be easier to read and that errors are fewer.

Q. We have received a few proposal s on missing information in the manuscript, and wishes for clarifications. Thus, we have done our best to clarify some more results, and to clarify how we have interpreted the results presented. These changes are now given as red text in the manuscript.

More specific comments:

­It is not stated what concentration/volumes of N3 adjuvant and pFliC were used in the vaccine formulations.

Answer: The N3 adjuvant was used in a 1% concentration and the FliC-DNA plasmid as 10 ug dose (6 uL/nostril).

­ELISA: the antigens used for coating are not adequately described. It seems that inactivated virus was used?

Answer: Yes, inactivated virus was coated at a 5 ug/mL concentration as evaluated by a checker-board titration ELISA analysis with known positive and negative control serum prior to be used.

We hope that these now performed changes and modifications of the manuscript fit the requests of the reviewer and that the manuscript may now be accepted for publication in the Vaccines journal.

Reviewer 2 Report

The manuscript describes vaccination in mice with WIV either in combination with the N3 lipid adjuvant, a DNA encoded TLR5 agonist secreted flagellin (pFliC), or both N3 and pFliC. While the combined use of N3/pFliC does not increase antibody levels in sera above that of N3 alone, it is quite striking that the inclusion of pFliC seems to polarize immune responses towards cellular immunity. The polarization is confirmed both with observations of IgG2a in sera, and ELISpot analyses of responses before and after a viral influenza challenge. Importantly, the combined use of N3/pFliC also increases survival in mice after heterologous H1N1 challenge.

Overall, the experiments are designed convincingly and in a manner that allows for conclusions to be drawn. That said, it is not clear which new information is offered by the study. It is already well known that the inclusion of an adjuvant can promote formation of cellular immunity after vaccination with WIV, and in fact this is likely what caused the unfortunate association between the 2009 influenza vaccine and narcolepsia in Scandinavia. A difference is here that the vaccine is given intranasally, but it would nevertheless be wise to discuss the present vaccine in relation to previous data on adjuvants promoting cellular immunity after WIV vaccinations.

While the text is fairly understandable, it would be beneficial to re-write the manuscript for increased clarity in descriptions of data, and the rationales behind the choices made. It would also strengthen the result section if some sentences were included to summarize the conclusions drawn for each experiment more clearly, and also with some level of comparison between the different figures. As an example, there is no explanation or discussion (or even a statement acknowledging) the fact that there is a discrepency between the HI assay and the ELISA results.

A re-writing of the manuscript would substantially strengthen the work, as would also a more detailed examination of which influenza antigens the vaccines actually induced immune responses against. Apart from the ELISpot assay where NP peptides were examined alongside influenza particles, immune responses were generally measured against whole virus particles. It would be interesting to know if the adjuvant could raise more antibody responses against neuraminidase in addition to hemagglutinin, and also if a significant portion of the vaccine induced antibodies were against internal influenza proteins. If these data were present, a more detailed explanation for the increased survival observed after vaccination with WIV+N3/pFliC could likely also be made. 

More specific comments:

-It is not stated what concentration/volumes of N3 adjuvant and pFliC were used in the vaccine formulations. 

-ELISA: the antigens used for coating are not adequately described. It seems that inactivated virus was used?

-ELISA: I would like a justification for why OD values between two different ELISA assays (IgG1 and IgG2a) can be used directly to calculate the IgG1/IgG2a ratio.

-Would it be possible to make the results in Table 2 more easily accessible by replacing the tabular format with graphic display? 

-HI assay: the definition of an HI titre of 40 as protective in M&M should also be reflected in Figure 3. 

-Line 223: As no significant difference was observed for vaccinations with WIV+N3 and WIV+N3/pFliC, it is difficult to see how N3 contributes to the effect of pFliC. This lack of significant difference between use of N3 or N3+pFliC is puzzling considering the differences observed in the Figures 2 and 3, and must be more stringently described.

-The subclass distribution of IgG responses in lung washes (Figure 2) should be shown, as these results could also support the Th1 polarization observed in sera after vaccinations with WIV and N3/pFliC.

-The added effect of N3+FliC is quite striking in Figures 2 and 3, but is not sufficiently described.

-Line 295: It should briefly be stated in an introductory sentence why IL-5 levels were evaluated.

-Line 310: Was there really a decrease in IL-5 production after vaccination with WIV+N3/pFliC? Decreased compared to what?

-Line 312: The time difference of 48 or 72 h should also be taken into consideration when comparing results in NMRI and C57BL/6 mice. 

-Figure 5 nicely confirms the Th1 polarization of CD4+ T cells, as also observed in mouse sera after vaccination with WIV+N3/pFliC, and also in 5D indicate a significant induction of CD8+ T cells. Given the good T cell responses induced against NP peptides, it is also not surprising that T cell responses are also observed against H3N2 in Figure 6. Perhaps provide a more detailed comparison of these results in the Results section?

-Lines 384 and 387: Days 180 and 270 post challenge or vaccination?

-Figure 8: Splenocytes were collected at what day after viral challenge?

-Table 3: It should be stated which day the animals were sacrificed at, as this will strongly influence serum levels. 

Figure 10: Lung washes collected at what day after challenge?

-Lines 500 to 514, while you could not perform a challenge with the homologue influenza strain, you did perform a challenge with heterologous H1N1. This should be discussed in relation to this section, because now it reads as though you indeed performed no influenza challenge. 

Author Response

We have now done our best to respond on the comments and questions provided by the reviewers of our manuscript entiteled: “Long-Lasting Mucosal and Systemic Immunity Against Influenza a Virus Is Significantly Prolonged and Protective by Nasal Whole Influenza Immunization with Mucosal Adjuvant N3 and DNA-Plasmid Expressing Flagellin in Aging In- and Outbred Mice.

We here below provide our responses, point-by-point and enclose the now re-written and language corrected manuscript. Our changes in the text are indicated with red text.

Reply on Reviewer questions and comments.

We thank reviewer 2 for valuable and insightful comments.

Q. We thank the reviewer 2 for the suggested idea that the manuscript could include a discussion regarding the use of influenza vaccination, adjuvants and side effects such as narcolepsia.

Answer: We have now included text concerning these issues in the Discussion section.

Q. Re-write with clearer aims or rationales for each activity. For instance a discussion on the discrepancy between ELISA and HAI should be discussed.

Answer: We have now added a paragraph concerning the value of functional antibody properties and ELISA-binding results detected.

Q. Interesting to know if the vaccination raises more or elevated immune responses against neuraminidase ? Also if the immunization raises more antibodies against the internal proteins ?

Answer: Unfortunately, we have not been able to get access to neuraminidase-antigens or internal influenza proteins in such an amount that we have been able to perform meaningful serological analyses.  Only, small amounts of recombinant hemagglutining protein representing the challenge virus was available and used to test for instance the lung-immunoglobulin reactivity against rHA. These data have now been included in the lung-IgG subclass results.

More specific comments:

­It is not stated what concentration/volumes of N3 adjuvant and pFliC were used in the vaccine formulations.

Answer: The N3 adjuvant was used in a 1% concentration and the FliC-DNA plasmid as 10 ug dose (6 uL/nostril).

­ELISA: the antigens used for coating are not adequately described. It seems that inactivated virus was used?

Answer: Yes, inactivated virus was coated at a 5 ug/mL concentration as evaluated by a checker-board titration ELISA analysis with known positive and negative control serum prior to be used.

­ELISA: I would like a justification for why OD values between two different ELISA assays (IgG1 and IgG2a) can be used directly to calculate the IgG1/IgG2a ratio.

Answer: In our hands, the use of serum IgG titers for calculating the IgG1/IgG2 ratios provides much greater inter-assay variation in results. This is probably due to the common phenomena that an upto 4-fold titer variation (upto 400%) is considered normal in biologically active sera. Instead, by chosing a certain serum dilution (1/100 or 1/1000) for comparison we seldom see a OD490-variation that is even two-fold or doubling of OD values (most often<200%). Thus, we feel that the later method is more robust and show less variability, something that is often appreciated in immunoassays.

­Would it be possible to make the results in Table 2 more easily accessible by replacing the tabular format with graphic display?

Answer: We will provide a graph to allow comparison with the Table 2.

­Q. HI assay: the definition of an HI titre of 40 as protective in M&M should also be reflected in Figure 3.

Answer: We have indicated in Figure three that the HI titer 40 can be suggested as a protective titer in human influenza vaccine trials (at least in younger individuals). In elderly individuals these titers are often to low to be well-correlated with protection from disease, as described in influenza vaccine studies in individuals aged >65 years of age. We have now added more information concerning this below figure 5 in Results and in the discussion.

­Q. Line 223: As no significant difference was observed for vaccinations with WIV+N3 and WIV+N3/pFliC, it is difficult to see how N3 contributes to the effect of pFliC. This lack of significant difference between use of N3 or N3+pFliC is puzzling considering the differences observed in the Figures 2 and 3, and must be more stringently described.

Answer: It is true as the reviewer no.2 point out, the total influenza-antigen-binding serum IgG ELISA titer changes against the tested influenza strains did not differ between animals receiving the WIV vaccine with N3 adjuvant or WIV vaccine with N3+FliC-DNA. To some degree the explanation for this lack of serum titer difference may depend on the strength of general influenza antigen-binding IgG/IgA detected in the used ELISA assay (see Figure 1). A great number of different influenza A antigens will be present on these plates, so we cannot distinguish between reactivity against the hemagglutinin (HA), neuraminidase (NA), nucleoprotein (NP), matrix proteins M1 and M2 or polymerase antigens. Thus, these serum titers just indicate a humoral immune response development against the influenza virus proteins, and differences between differently immunized groups. Therefore, the serological results obtained in Figures 2 and 3 better identify the local lung virus-binding antibody responses (Figure 2) and the functional hemagglutination inhibiting serum titers (Figure 3).  

­Q. The subclass distribution of IgG responses in lung washes (Figure 2) should be shown, as these results could also support the Th1 polarization observed in sera after vaccinations with WIV and N3/pFliC.

Answer: We thank the reviewer for this proposal. Results from ELISA measuring IgG subclasses in lung washes have now been added as a new Figure 4. It seems like the lung IgG subclass pattern reflect the pattern seen in serum.

­Q. The added effect of N3+FliC is quite striking in Figures 2 and 3, but is not sufficiently described.

Answer: We have added further descriptions of Figures 2 and 3 effects by FliC-DNA in N3. The serological results obtained in Figures 2 and 3 most likely identify at higher degree the local lung virus-binding antibody responses (Figure 2) and the functional, potentialy virus neutralizing hemagglutination inhibiting serum titers (Figure 3), and the differences between study groups.  In these two figures, obtained data (Figure 2) clearly indicate that the combined effect of adjuvants N3 plus FliC-DNA significantly enhance the local lung IgA and IgG levels against influenza antigens compared with N3 or FliC alone. Also, the functional anti-viral HI titers were more elevated over time (90 days after immunization start, in Figure 3) in animals receiving N3 plus FliC-DNA adjuvant, instead of only one adjuvant separately.

­Q. Line 295: It should briefly be stated in an introductory sentence why IL­5 levels were evaluated.

Answer: In our experience, the mIL-5 cytokine is a more robust cytokine than the more short-lived mIL-4. In our hands the mIL-4 cytokine is rapidly degraded or consumed, so we have some difficulties to monitor it with our available assays. According to the literature we think that at a high degree the mIL-5 cytokine levels well represent the humoral (or Th2-type) immune responsiveness as a good supplement to the mIL-4 cytokine. This cytokine have furthermore a better association with influenza B cell reactivity in elderly individuals (we have enclosed a new reference article concerning this matter.

­Q. Line 310: Was there really a decrease in IL­5 production after vaccination with WIV+N3/pFliC? Decreased compared to what?

Answer: We thank the reviewer for this comment. We agree that there is no decrease in IL-5 secretion, but instead a lower level of IL-5 detected in comparison with the other adjuvants only among the NMRI mice (Figure 6A). We have now corrected this in the text below figure 6.

­Q. Line 312: The time difference of 48 or 72 h should also be taken into consideration when comparing results in NMRI and C57BL/6 mice.

Answer: The differences between the peak levels of cytokines measured between the two mouse strains studied resulted in the different time-points used for the animals. In pre-evaluations studies we performed time-kinetic analyses that suggested us to choose the different time points. We have not included these control analyses, and time kinetics, but we will mention them in the Materials and Methods section.

­Q. Figure 5 nicely confirms the Th1 polarization of CD4+ T cells, as also observed in mouse sera after vaccination with WIV+N3/pFliC, and also in 5D indicate a significant induction of CD8+ T cells. Given the good T cell responses induced against NP peptides, it is also not surprising that T cell responses are also observed against H3N2 in Figure 6. Perhaps provide a more detailed comparison of these results in the Results section?

Answer: Figure 5, ELIspot analyses revealed that the use of adjuvants made a significant difference in the cell-mediated T cell response patterns, but at different degree in the two strains of mice. One need to bear in mind that these studies were performed with single, previously identified strain-specific CD8+ T cell epitopes only.

The results in Figure 5 thus suggest the following. After the initial immunization

­Q. Lines 384 and 387: Days 180 and 270 post challenge or vaccination? ­Figure 8: Splenocytes were collected at what day after viral challenge?

Answer: This figure is now changed to Figure 9A-D. The days should represent Days post-final immunization. We have clarified this in the figure ledgend.

Figure 8 is now changed to Figure 10: The spleens were collected at the day of sacrifice (Day 15 to 30).

­Q. Table 3: It should be stated which day the animals were sacrificed at, as this will strongly influence serum levels.

Answer: We have now stated the days when the mice were sampled in the Figure ledgend.

Q. Figure 10: Lung washes collected at what day after challenge?

Answer: At the day of sacrifice. (In group  WIV no adjuvant during day 6 – 15, in group WIV + N3 at day 12 – 21 and in group WIV N3 plus FliC-DNA at day 30 (180 days post challenge).

­Lines 500 to 514, while you could not perform a challenge with the homologue influenza strain, you did perform a challenge with heterologous H1N1. This should be discussed in relation to this section, because now it reads as though you indeed performed no influenza challenge.

Answer: Yes, we apologize for this unclear presentation. What we aimed at was to say that we were unable to perform homologous challenge, due to lack of such pathogenic challenge virus. So, we had to perform heterologous challenge instead. This is now clearer added into the text.

Round  2

Reviewer 2 Report

Thank you for nicely responding to my previous comments! I find that the paper has been much improved. That said, I still believe that the text and wording still need some more refinement. My suggestion would be to go through the full paper, and again check that all the sections and sentences clearly convey your message. Please find below some examples of sentences that particularly need to be revised for increased readability:

Line 55-57: Please revise the sentence for clarity.

Line 57-59: Would improve the sentence if you remove "usually in mice and ferrets as well as in clinical phase I studies". Alternatively, please revise the sentence for clarity.

Line 81: Please reverse wording of "speed Production" to "Production speed". 

Line 85: Please change to "...additon of lipid.."

Line 90-92: Please revise sentence for better wording.

Line 93-96: It is really hard to understand this sentence, so please revise. 

Line 235-236: As far as I can see, there is not a significant difference between WIW/FliC/N3 and WIW/N3. While the trend is the same in both NMRI and C57BL/6, it is not significant. This should also be reflected in the text.

Line 244: Do like Figure 2. Please remember to refer to this figure also in the text...

Line 290: You should also spend a few sentences in the text describing observations from Figure 4, and also reference the figure in the text. 

Line 325: I still think that you should also include a sentence in the text describing why IL5 was chosen. 

Line 632: Please exchange "roll" for "role"

Line 662: You should probably remove the text instruction. 

Author Response

We thank the reviewer for the valuable comments, proposals and requests for clarifications in the second round of review. We have here below added our replies on the comments, as point-by-point answers.

Vaccines-486769 Reviewer reformulation requests:

Q1. Line 55-57: Please revise the sentence for clarity.

Answer: Line 55-57: We thank the reviewer for this proposal and request for clarifications of the text. We have below added our text modification in the suggested text lines, and added it into the manuscript.

Current sentence: Development of influenza vaccines capable of inducing a broader immunity than what the seasonal subunit HA/NA or whole influenza vaccines have generated great interest due to the regularly arriving new epidemic outbreaks

Our new proposal, is now added into the manuscript as follows: It would be highly desirable to develop influenza vaccines that provide broader influenza-specific immune responses than what can be obtained with the currently available commercial inactivated flu-vaccines. If stronger and more long-lasting cell-mediated and humoral flu-specific immunity could be obtained, it would be more likely that the obtained immunity could better protect against disease, in future epidemics.

Q2. Line 57-59: Would improve the sentence if you remove "usually in mice and ferrets as well as in clinical phase I studies". Alternatively, please revise the sentence for clarity.

Answer: We thank the reviewer for this valuable advice. We have changed the text accordingly, and deleted the text suggested.

Q3. Line 81: Please reverse wording of "speed Production" to "Production speed". 

Answer: We have now changed the text on line 81 as suggested.

Q4. Line 85: Please change to "...additon of lipid.."

Answer: We have changed the text as suggested on line 85.

Q5. Line 90-92: Please revise sentence for better wording.

Answer: We thank the reviewer for this text modification request. We hope that the below added text will be better and clearer.

Line 90-92: Current sentence: It also drives the recruitment of CD103+ DCs in the mesenteric lymph nodes as well as switching of flagellin-specific B cells to IgA and drives mucosal T and B cell responses [21].

New proposal added into the manuscript: In previous studies, the presence and uptake of bacterial flagellin proteins by CD103+ dendritic cells (DC) have resulted in their increased presence in mesenteric lymphnodes. Further, the flagellin-proteins have been shown to increase B-cells to switsch into IgA secreting cells, thereby enhancing the mucosal B and T cell responses against flagellin (21)

Q6. Line 93-96: It is really hard to understand this sentence, so please revise. 

Answer: We thank the reviewer for this text modification request. We hope that the below added text will easier to understand and clearer.

Line 93 – 96: Current sentence: Until now, no influenza vaccination using inactivated whole influenza A virus (WIV) as nasally given immunogen where the adjuvant an endogenously secreted, DNA-plasmid expressed TLR5-agonistic with de-glycosylated flagellin C S.Typhimurium-derived adjuvant was used and compared with and without a cationic lipid-based N3 emulsion aimed for mucosal delivery.

New proposal, added into the manuscript: The novelty of the present vaccination design of inactivated influenza A virus is the combination of previously never studied combined adjuvants. Thus, the adjuvants, the cationic lipid N3 alone or N3 lipid mixed with DNA-plasmid expressing the TLR5-agonistic, de-glycosylated flagellin C-protein (S.Typhimurium-derived) mixed with the WIV/Salomon Island A/H1N1 2007-antigen prepared as an emulsion for nasal mucosal administration.

Q7. Line 235-236: As far as I can see, there is not a significant difference between WIW/FliC/N3 and WIW/N3. While the trend is the same in both NMRI and C57BL/6, it is not significant. This should also be reflected in the text.

Line 244: Do like Figure 2. Please remember to refer to this figure also in the text...

Answer: We thank the reviewer for this observation, and we have now added more information and clarifications pointing out the lack of significant differences. New text added: Thus, the humoral influenza-specific immune responses between mice receiving WIV with N3 or N3 and FliC-DNA were not significantly different, when analyzed by ELISA. Thus, binding antibodies alone, may provide a misleading immune pattern from a functional point of view, as can be seen in the more detailed assays such as subclass IgG ELISA (Figures 2, 3 and 4) or functional antiviral assays such as in virus-inhibition assays (Figure 5).

Q8. Line 290: You should also spend a few sentences in the text describing observations from Figure 4, and also reference the figure in the text. 

Answer: We thank the reviewer for observing this and for this request to comment Figure 4. We have now added a text immediately below the Figure 4, with figure reference.  New text added: The subclass IgG pattern seen in lung washes collected after the booster immunization (Figure 4) show a significantly different pattern in the animals immunized nasally with WIV with N3 and FliC (-gly) DNA that seen in animals receiving WIV or WIV/N3 adjuvant. In the N3/FliC-DNA groups of both outbred NMRI and inbred C57BL/6 a significantly stronger influenza-specific IgG2-response was detectable in lung washes suggesting a more balanced Th2/Th1 immunresponse against the H1 hemagglutinine antigen.

Q9. Line 325: I still think that you should also include a sentence in the text describing why IL5 was chosen.

Answer: We thank the reviewer for this request for additional comments on the selection of IL-5 analyses. We have now added the following text with our comments on the IL-5 detection and quantification in the Figure 6 legend.  “The different time points chosen for the two mouse strains were determined in an in vitro pre-study influenza/ConA stimulation of spleen cells, and the optimal time point for highest levels of IL-5 secretion in spleen cells of adjuvant-vaccinated aged animals was chosen (data not shown).”

Also in Materials and Methods in paragraph 2.6 we have added the following information concerning IL-5: The different time points chosen for the two mouse strains were determined in an in vitro pre-study influenza/ConA stimulation of spleen cells, and the optimal time point for highest levels of IL-5 secretion in elderly animals was chosen (data not shown). Furthermore, as shown by McDonald et al. 2017, IL-5 secretion was shown to be secreted at higher amounts for a longer period than IL-4 in vitro (or possibly consumed less rapidly) in aged mice, thus making IL-5 easier to use as a Th2-biomarker than IL-4.

Q10. Line 632: Please exchange "roll" for "role"

Answer: We thank the reviewer for observing this error. We have now performed the change as suggested.

Q11. Line 662: You should probably remove the text instruction. 

Answer: We thank the reviewer for observing this error. We have now performed the change as suggested.
